# A Two-Character Change in Transformer Architecture Promotes Ideal Token Geometry

## Abstract

We hypothesize that in the optimal geometric configuration of token embeddings for transformer classifiers, tokens should collapse to single points according to their classes, and these points themselves should exhibit Neural Collapse. We study whether current transformers achieve this configuration through principal component projections, cosine similarity measurements, analysis of variance on token embeddings, and Neural Collapse measurements and find that they fall far short of the conjectured ideal. To address this, we introduce a simple modification to attention that brings token embeddings markedly closer to the conjectured configuration and yields consistent performance improvements across benchmarks.

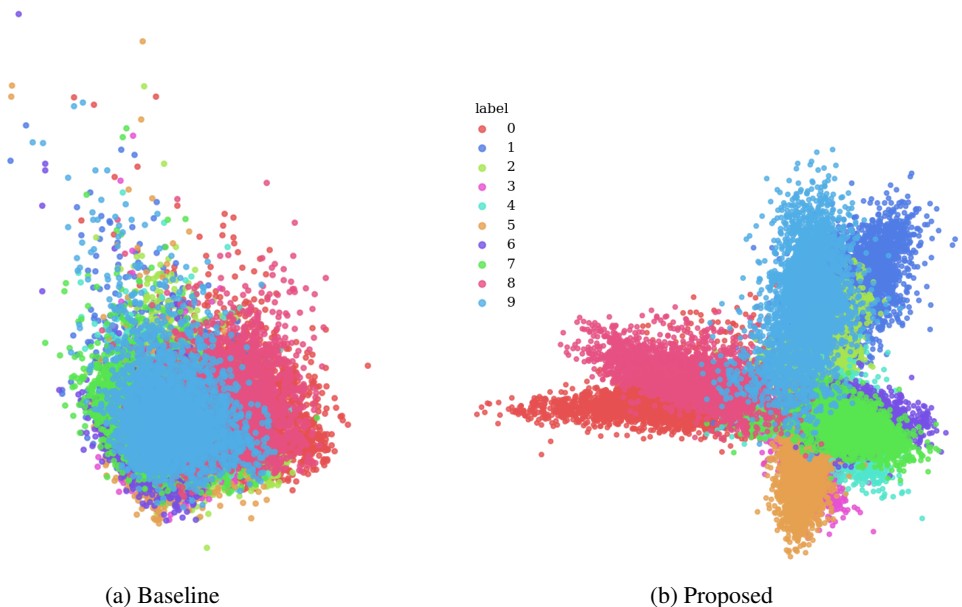

(a) Baseline          (b) Proposed

Figure 1: Proposed modification cleanly separates tokens into clusters according to class.

## 1 Introduction

### 1.1 Motivation and Problem Statement

The transformer architecture has achieved remarkable success across a wide range of tasks since its introduction by Vaswani et al. (2017). From image classification to language modeling, many of these tasks reduce to classification problems in which the transformer learns input-token embeddings and then applies a final-layer classifier to predict the class of the input sequence.

In the context of deep networks, Papyan et al. (2020) showed that as training approaches zero error, the final-layer representations of standard classifiers exhibit a distinctive geometry—termed **Neural Collapse (NC)**—with four defining properties:

**NC1** Final-layer representations collapse to their class means.

**NC2** The class means converge to a simplex equiangular tight frame (ETF).

**NC3** The linear classifier weights align with the corresponding class means.

**NC4** The classifier predicts each class via nearest-class-mean decision boundaries.

While these phenomena have been extensively studied in convolutional networks for image classification (Papyan et al., 2020; Han et al., 2022), transformers present a fundamentally different setting: inputs are tokenized sequences, and the learned representations consist of token embeddings rather than a single feature vector. This naturally raises the following questions:

**Q1:** *What is the ideal geometric configuration of token embeddings for classification?*

**Q2:** *Do current transformers achieve this configuration?*

**Q3:** *If not, can architectural modifications induce it?*

At its core, the transformer consists of a sequence of attention layers (Vaswani et al., 2017). Several theoretical works show that attention leads to **rank collapse**, where token embeddings converge to a single vector (Dong et al., 2023; Noci et al., 2022; Geshkovski et al., 2024). These results indicate an inherent inductive bias of attention toward collapsing tokens.

## 1.2 CONTRIBUTIONS

To address Question Q1, we draw on insights from NC and prior work on rank collapse to introduce Neural Token Collapse (NTC), our hypothesized ideal geometry for token embeddings in classification tasks. NTC is defined formally in Section 3 and characterized by three core properties:

**NTC0** Tokens within the same sequence collapse to a single vector.

**NTC1** All sequences of the same class collapse to the class mean.

**NTC2-4** The class means and classifier weights satisfy NC2, NC3 and NC4.

To address Question Q2, in Section 4, we assess to what extent the token embeddings of current transformers conform to NTC by applying principal component analysis, cosine similarity, analysis of variance, and NC measurements. Our findings show that standard transformers fail to achieve NTC.

Finally, to address Question Q3, we introduce in Section 2 a simple architectural modification designed to overcome this limitation. It is motivated by the observation that standard transformers uses attention to compute a contextual mean and adds it to each token, increasing the tokens' mean relative to their variance. Layer normalization then projects the tokens back onto a sphere, causing them to cluster more closely together. This process is inefficient because it introduces norm-changing radial movements that are subsequently removed by layer normalization (Figure 2). Our modification moves the tokens directly along their variance directions, so the tokens can collapse without relying on the intermediate mean shift and projection (Figure 2). As shown in Section 4, this approach promotes NTC more effectively and consistently improves performance across multiple datasets.

## 2 PROPOSED ARCHITECTURE MODIFICATION

In this section, we propose our modification to the transformer architecture, which we motivate by analyzing how attention affects the geometry of token embeddings.

### 2.1 STANDARD ATTENTION MECHANISM

Consider a sequence of $T$ token embeddings $(x_1, \ldots, x_T)$ with each $x_i \in \mathbb{R}^d$, collected into a matrix $X \in \mathbb{R}^{T \times d}$. Attention (**Attn**) projects $X$ into queries, keys, and values using trainable matrices $W_Q, W_K, W_V \in \mathbb{R}^{d \times d_k}$:

$$Q = XW_Q, \qquad K = XW_K, \qquad V = XW_V.$$

It then computes scaled dot-product weights and outputs a weighted sum of the values:

$$P = \text{softmax}\left(\frac{QK^\top}{\sqrt{d_k}}\right), \quad \mathbf{Attn}(X) = PV. \tag{1}$$

Building on this definition, multi-head attention (**MHA**) computes $h$ attention heads in parallel, concatenates their outputs, then applies a trainable projection matrix $W_o \in \mathbb{R}^{hd_k \times d}$. The result is added back to the input via a residual connection (He et al., 2015):

$$X' = X + \mathbf{MHA}(X). \tag{2}$$

After this residual update, the token embeddings are passed through LayerNorm (Ba et al., 2016), which projects the tokens back onto a sphere with radius $\sqrt{d}$. This process repeats at each layer.

**Interpretation.** Equation 2 increases the tokens' mean relative to their variance, so the subsequent renormalization draws them closer together on the sphere. Figure 2 illustrates this process: token embeddings $\vec{x}_1, \vec{x}_2$ (black arrows) are shifted by their attention-weighted means $(PX)_1, (PX)_2$ (blue arrows) along the orange arrows, and after projection they end up closer together on the sphere (green points).

## 2.2 THE TWO-CHARACTER CHANGE

We modify the standard attention mechanism into a *Laplacian mechanism*, denoted by $\mathcal{L}$:

$$P = \text{softmax}\left(\frac{QK^\top}{\sqrt{d_k}}\right), \qquad \mathcal{L}(X) = V - PV. \tag{3}$$

This modification amounts to adding two extra characters, "$-$" and "$V$", hence the title. The term "Laplacian" will be justified in Section 5.

**Remark.** *It is important to emphasize that this modification differs from the standard skip connection applied around* $\mathbf{Attn}$ *(or* $\mathbf{MHA}$*), as the residual stream carries an additional* $XW_v$ *term.*

**Interpretation.** While $\mathbf{Attn}$ computes the mean of tokens, $\mathcal{L}$ computes the difference between tokens and the mean. If every attention head is replaced with $\mathcal{L}$, then Equation 2 allows each token to move directly toward the mean. This eliminates the redundant radial movement later canceled by projection. Figure 2 shows this effect: token embeddings $\vec{x}_1, \vec{x}_2$ (black arrows) move directly towards the means $(PV)_1, (PV)_2$ (blue arrows) along the variance direction (orange arrows). The green points are the projected tokens, which end up much closer together with the projection step playing only a minor role. We provide empirical evidence for this geometric interpretation in Appendix D.

## 2.3 MIXING BOTH MECHANISMS

If one is willing to change more than two characters, then the mechanism can be extended further by mixing $\mathbf{Attn}$ and $\mathcal{L}$ in transformers. Specifically, we propose using $\mathcal{L}$ for some heads and $\mathbf{Attn}$ for others. There are many possible ways to do this and we find the following two simple strategies to be effective:

1. Fix $m < h$. In each layer, assign $\mathbf{Attn}$ to $m$ heads and $\mathcal{L}$ to the remaining $h - m$ heads.
2. Assign $\mathbf{Attn}$ to all heads in the first half of the transformer blocks and $\mathcal{L}$ to all heads in the second half.

Both strategies are straightforward to implement and introduce no additional trainable parameters. We describe more mixing strategies we have tried in Appendix E.

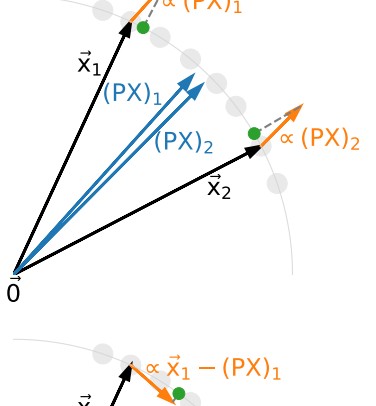

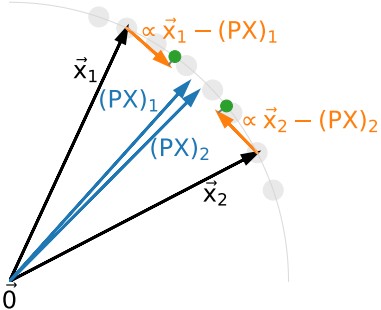

Figure 2: Baseline (top) and proposed (bottom)

**Interpretation.** This setup allows some heads to move tokens toward their mean while others can move tokens along their variance direction, giving the model greater flexibility to steer tokens toward the ideal geometry.

## 3 IDEAL TOKEN GEOMETRY

Before formally defining our conjectured ideal token geometry, we need to define several quantities that describe the distribution of token embeddings. We consider a dataset $D$ with $N$ data points and $C$ classes, where class $c$ contains $N_c$ data points so that $N = \sum_{c=1}^{C} N_c$. Each data point is tokenized into a sequence of $T$ tokens. For indices $1 \leq c \leq C$, $1 \leq i \leq N_c$, and $1 \leq t \leq T$, let $X_{t,i,c}$ denote the embedding of the $t$-th token of the $i$-th data point in class $c$.

### 3.1 TOKEN MEANS

We first define several means. For each data point we compute the *sequence token mean*, which is the average of its $T$ token embeddings:

$$\mu_{i,c} = \mathbf{Ave}_{t} X_{t,i,c}.$$

For each class, we compute the *class token mean*, obtained by averaging the sequence means over all $N_c$ data points in that class:

$$\mu_c = \mathbf{Ave}_{i} \mu_{i,c} = \mathbf{Ave}_{t,i} X_{t,i,c}.$$

Finally, we compute the *global token mean* by averaging the class means over all $C$ classes:

$$\mu_G = \mathbf{Ave}_{c} \mu_c = \mathbf{Ave}_{t,i,c} X_{t,i,c}.$$

### 3.2 TOKEN VARIANCES

Using $\mu_{i,c}$, $\mu_c$, and $\mu_G$, we define several measures of variance that quantify different aspects of token distribution.

The *within-sequence variance* measures how far individual token embeddings of a single data point deviate from the token mean of that data point itself on avearge:

$$\text{WithinSeqVar} = \mathbf{Ave}_{t,i,c} \left\| X_{t,i,c} - \mu_{i,c} \right\|^2.$$

The *within-class variance* measures how far each sequence's token mean deviates from the corresponding class token mean on average:

$$\text{WithinClassVar} = \mathbf{Ave}_{i,c} \left\| \mu_{i,c} - \mu_c \right\|^2.$$

The *between-class variance* measures how far each class token mean deviates from the global token mean:

$$\text{BetweenClassVar} = \mathbf{Ave}_{c} \left\| \mu_c - \mu_G \right\|^2.$$

Finally, the *total variance* quantifies the overall variance of all token embeddings around the global token mean:

$$\text{TotalVar} = \mathbf{Ave}_{t,i,c} \left\| X_{t,i,c} - \mu_G \right\|^2.$$

### 3.3 HIERARCHICAL VARIANCE DECOMPOSITION

By expanding the squared norm, the total variance decomposes additively into the three components above:

$$\text{TotalVar} = \text{BetweenClassVar} + \text{WithinClassVar} + \text{WithinSeqVar} .$$

This decomposition mirrors an ANOVA-style breakdown of the total energy into within-sequence, within-class, and between-class contributions.

### 3.4 NEURAL TOKEN COLLAPSE

Using the means and variances defined in the previous section, we formally define *Neural Token Collapse* (NTC), our conjectured ideal token geometry for classification. This regime is characterized by the following properties:

**NTC0** All tokens within a sequence coincide with that sequence's mean,

$$X_{t,i,c} = \mu_{i,c} \qquad \forall\, 1 \le c \le C,\ 1 \le t \le T,\ 1 \le i \le N_c,$$

which implies
$$\mathrm{WithinSeqVar} = 0.$$

**NTC1** All sequence means within a class coincide with the class mean,

$$\mu_{i,c} = \mu_c \qquad \forall\, 1 \le i \le N_c,$$

which implies
$$\mathrm{WithinClassVar} = 0.$$

Combined with the token-level collapse above, this further implies
$$\mathrm{TotalVar} = \mathrm{BetweenClassVar}.$$

**NTC2–4** The class means
$$\{\mu_c : 1 \le c \le C\},$$

together with the classifier weights, satisfy NC2, NC3, and NC4 as defined in Papyan et al. (2020).

## 4 EXPERIMENTS

### 4.1 SETUP

Our experiments are based on the DeiT-3 vision transformer (Touvron et al., 2022), which is widely considered a strong baseline for image classification. By default, we applied the modification described in Section 2.3 to the ViT-B model. For strategy 1, we focus mainly on three options: $m = 0, 1, 3$. Note that $m = 0$ is equivalent to using the Laplacian mechanism for all heads, and we discuss the impact of different values of $m$ in Section 4.6. Throughout the paper, we denote these three options as "ViT-B-0P", "ViT-B-1P" and "ViT-B-3P", respectively. We refer to the baseline vision transformer as "ViT-B" and to the approach described in strategy 2 as "ViT-B-Mix-Depth."

We trained these models on CIFAR-10, CIFAR-100 (Krizhevsky, 2009), and ImageNet-1k (Deng et al., 2009). For each dataset, all models shared the same training recipe. Full details of the training setup is provided in Appendix B.

In the following subsections, we use a range of metrics to analyze the token geometry of the models, assessing whether the proposed modifications promote NTC more relative to the ViT-B baseline.

### 4.2 PRINCIPAL COMPONENT ANALYSIS OF TOKEN EMBEDDINGS

Let $X$ denote a batch of $B$ sequences of token embeddings, represented as a tensor of shape $(B, T, d)$, where $T$ is the sequence length and $d$ is the embedding dimension. We apply principal component analysis (PCA) to $X$ and project the tokens onto $\mathbb{R}^2$ using the top-two principal components, as formally defined in Algorithm 1 in Appendix A. For visualization, tokens belonging to the same class are plotted in the same color.

Figure 1a illustrates the resulting two-dimensional PCA projection of the last-layer token embeddings for ViT-B and ViT-B-1P trained on CIFAR-10. For ViT-B, tokens from different classes overlap and exhibit no clear geometric structure. In contrast, the token embeddings of ViT-B-1P form well-separated clusters. The projections for ViT-B-0P, ViT-B-3P, and ViT-B-Mix-Depth display a pattern similar to that of ViT-B-1P (see Appendix C).

## 4.3 ANALYSIS OF VARIANCE (ANOVA) OF TOKEN EMBEDDINGS

Following the definitions in Section 3.2, we compute the total variance (TotalVar) of the token embeddings and its three components—between-class variance (BetweenClassVar), within-class variance (WithinClassVar), and within-sequence variance (WithinSeqVar)—each expressed as a fraction of TotalVar so that they sum to one.

Figure 3 presents these fractions for ViT-B and ViT-B-1P trained on CIFAR10. For ViT-B, WithinSeqVar constitutes the majority of the total variance, indicating a limited amount of token-level collapse. In contrast, ViT-B-1P substantially reduces WithinSeqVar while leaving WithinClassVar roughly unchanged, implying a much higher degree of token-level collapse. Moreover, variance energy shifts from WithinSeqVar to BetweenClassVar, suggesting that ViT-B-1P pushes the class means farther apart than ViT-B. These findings reinforce the patterns observed in the PCA visualization.

## 4.4 LAYERWISE NEURAL TOKEN COLLAPSE

We measure the average cosine similarity between all pairs of tokens within the same sequence, defined as

$$\frac{1}{B}\sum_{b=1}^{B}\frac{1}{T(T-1)}\sum_{\substack{i,j=1 \\ i \neq j}}^{T}\frac{\langle X_{b,i},\,X_{b,j}\rangle}{\|X_{b,i}\|\,\|X_{b,j}\|}\,,$$

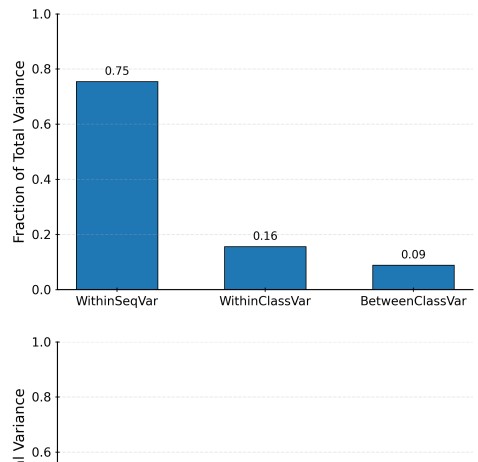

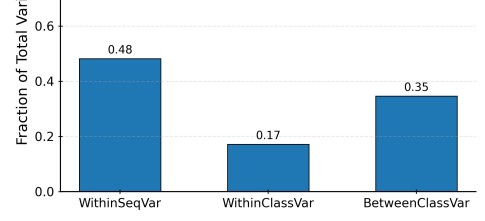

Figure 3: ViT-B-1P (bottom) moves variance from within-image to between-class compared to ViT-B (top). The total variance decomposes additively as TotalVar $=$ BetweenClassVar $+$ WithinClassVar $+$ WithinSeqVar.

where $X_{b,i} \in \mathbb{R}^d$ denotes the $i$-th token in the $b$-th sequence. This quantity, denoted CosSim, lies between $-1$ and $1$. A larger value indicate that, on average, tokens within the same sequence are more strongly aligned.

Figure 4 shows CosSim for the output tokens of each transformer block (after the final normalization layer) in models trained on ImageNet. For ViT-B, CosSim remains small across all layers, indicating limited token-level collapse. In contrast, all four proposed models exhibit a much steeper increase in CosSim, reaching substantially higher values in the deeper layers. This behavior suggests that the proposed models induce markedly stronger alignment among tokens than the ViT-B baseline.

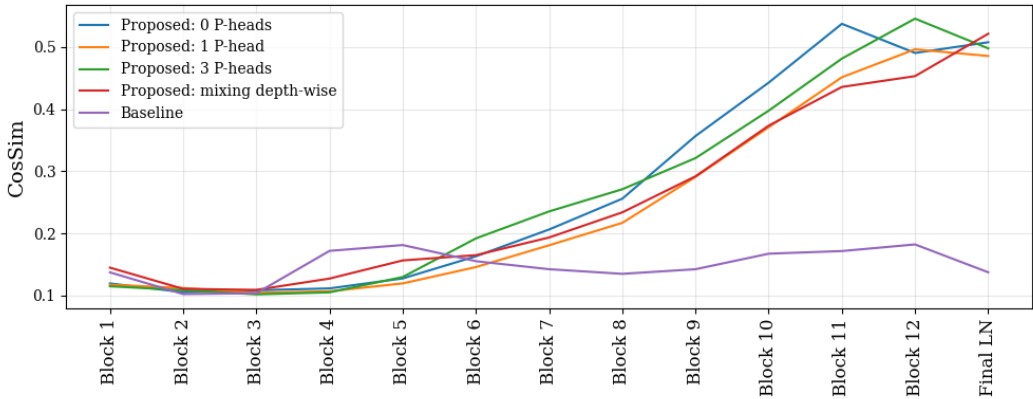

Figure 4: Proposed method induces a significantly steeper increase in CosSim across depth. Measured on ImageNet.

## 4.5 NEURAL COLLAPSE METRICS AND VISUALIZATION

We assess NC2–NC4 using the methodology of Han et al. (2022) and complement these metrics with a visualization technique introduced by Fisher et al. (2024) (Algorithm 2). Full details are provided in Appendix F.

Figure 5 compares the four NC metrics across the different models trained on CIFAR10. Figure 5a shows that the class means are more equinorm for all three proposed models than for the baseline. Figure 5b further indicates that all four proposed models produces class means that are more equiangular. Overall, these results suggest that our modifications better promote a simplex ETF structure in the class means. We observe no significant differences in self-duality or in the NC2 metrics for the classifier weights across models (Figures 5b, 5c). Finally, Figure 5d shows that the proposed models move significantly closer to an NCC classifier.

Figure 5e visualizes the token embeddings projected onto the classifier. While the baseline displays diffuse, overlapping clouds of tokens from different classes, the proposed modification produces well-separated clusters with a clear simplex-like arrangement. This pattern reinforces the NC2–NC4 metrics by illustrating directly that the modified models achieve stronger token-level collapse and more distinct class separation than the ViT-B baseline.

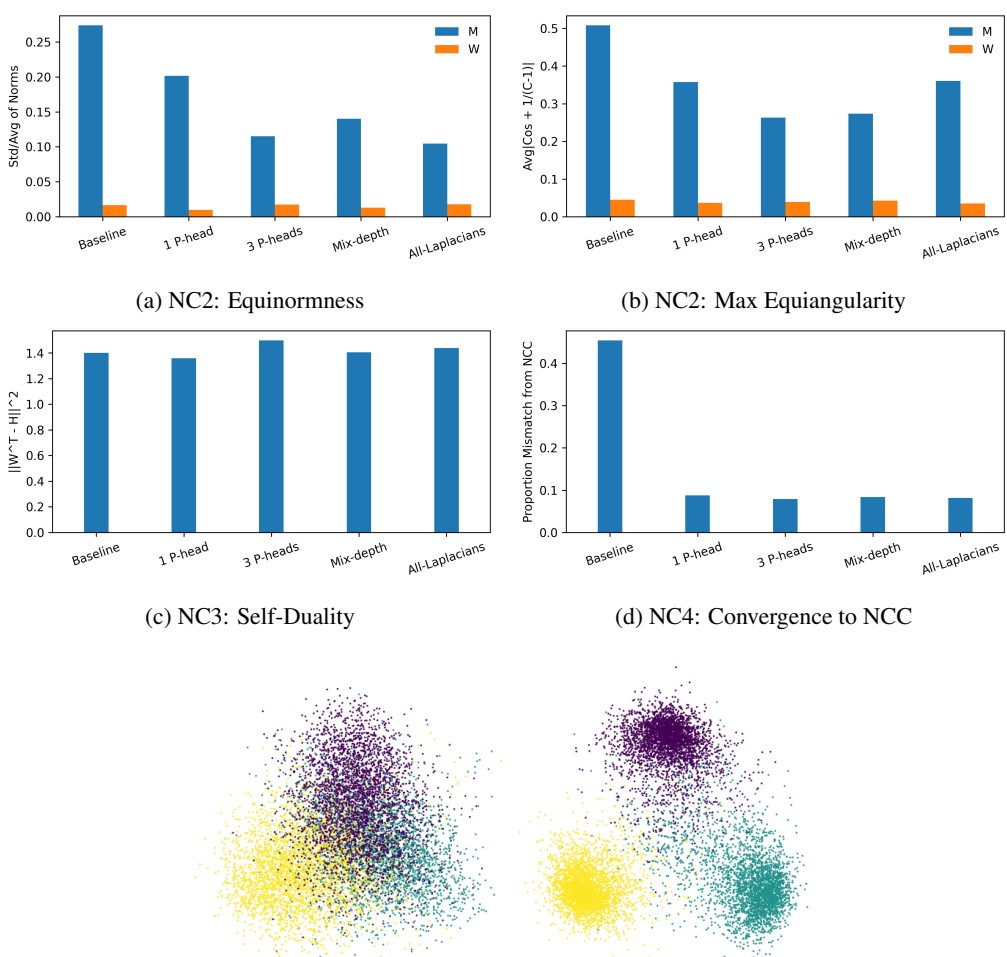

(a) NC2: Equinormness

(b) NC2: Max Equiangularity

(c) NC3: Self-Duality

(d) NC4: Convergence to NCC

(e) Projection of token embeddings onto the classifier. Baseline (left) versus proposed (right).

Figure 5: Neural-collapse metrics on CIFAR-10 (top) and projection of token embeddings onto the classifier (bottom). Here $M$ denotes the matrix of class means—averaged over all tokens and instances within each class—while $W$ denotes the classifier weight matrix.

## 4.6 PROPOSED METHOD IMPROVES PERFORMANCE

### 4.6.1 IMAGE CLASSIFICATION

Table 1 compares the top-1 test accuracies on multiple datasets between our proposal and the baseline. It clearly illustrates that the proposed models consistently produce meaningful improvements upon the baseline across all datasets. Most notably, the improvements on ImageNet-1k provide evidence that the proposed models work well on large-scale datasets. The significant improvements produced by ViT-B-1P and ViT-B-3P on CIFAR100 (4-5%) also suggest that the proposed models are more data-efficient as CIFAR100 has limited data points per class.

Table 1: Top–1 test accuracy (%) of models across datasets (mean $\pm$ standard deviation). The proposed models yield consistent performance improvement.

| Model | CIFAR-10 | CIFAR-100 | ImageNet-1k |
|---|---|---|---|
| Baseline (ViT-B) | 90.41$\pm$0.15 | 61.41$\pm$0.36 | 81.2 |
| Proposed (ViT-B-0P) | 91.74$\pm$0.10 | 65.39$\pm$0.07 | 82.02 |
| Proposed (ViT-B-1P) | **91.83$\pm$0.08** | **66.05$\pm$0.22** | **82.18** |
| Proposed (ViT-B-3P) | 91.83$\pm$0.10 | 65.44$\pm$0.36 | 82.17 |
| Proposed (ViT-B-Mix-Depth) | 91.79$\pm$0.18 | 61.55$\pm$0.39 | 82.07 |

Next, we compare the top-1 accuracies of models with different numbers $m$ of standard attention heads (which we refer to as $P$ heads). Since the baseline model ViT-B has 12 $P$ heads in each layer, we trained models with $m \in \{0, 1, 3, 6, 9, 12\}$. Figure 6 illustrates that while using the Laplacian heads only ($0P$) already induces noticeable improvement upon the baseline ($12P$), incorporating a small number of $P$ heads ($m = 1, 3$) yields further improvements on ImageNet-1k. Since this pattern is consistent across datasets (see Appendix E), we recommend using a small number of $P$ heads as the default strategy.

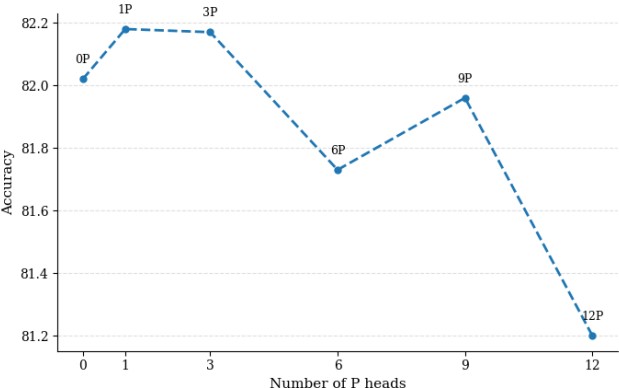

Figure 6: ImageNet accuracy as a function of the number of $P$ heads.

### 4.6.2 AUTOREGRESSIVE LANGUAGE MODELLING

We trained decoder-only transformer models for autoregressive next-token prediction to investigate the impact of the Laplacian mechanism on language modelling tasks. Specifically, we took a GPT-2 style transformer model with 836 million parameters as the baseline. Following Strategy 1, we assigned the Laplacian mechanism to 15 heads and standard attention to 5 heads in each layer. We trained the two models on 20 billion tokens from the FineWebEdu dataset (Lozhkov et al., 2024) and evaluated them on a variety of zero-shot downstream tasks. Full details of the experiments can be found in Appendix B.

Table 2: Zero-shot results of 836M GPT2-style decoder-only transformers on downstream datasets. "ACC" means accuracy (higher is better).

|  | ARC-Easy (ACC) | ARC-Challenge (ACC) | HellaSwag (ACC) | PIQA (ACC) | RACE (ACC) | OpenBookQA (ACC) | WinoGrande (ACC) | SciQ (ACC) |
|---|---|---|---|---|---|---|---|---|
| Baseline (836M) | 63.8 | 31.48 | 36.22 | 68.55 | 30.62 | 33.2 | 53.99 | 84.1 |
| Proposed: 5 P-heads (836M) | **64.56** | **32.08** | **36.4** | **68.71** | **32.15** | **34.6** | **54.78** | **85.2** |

Our results (Table 2) show that incorporating the Laplacian heads leads to better performance on various downstream datasets. This results corroborates several previous works on the token collapse phenomenon in autoregressive transformers, a connection we discuss in more detail in Section 6.

## 5 RELATION TO DIFFUSION OVER GRAPHS

Our Laplacian mechanism can be understood through the lens of diffusion on graphs, in particular the discrete heat equation.

**Connection to graphs**  Consider a sequence of $T$ tokens. Treat each token as a vertex in a graph $G = (V, E)$ with $|V| = T$. The attention weights from equation 1,

$$P = \text{softmax}\Big(\frac{QK^\top}{\sqrt{d_k}}\Big) \in \mathbb{R}^{T \times T},$$

define directed edge weights: $P_{ij}$ encodes the normalized influence of token $j$ on token $i$. Because each row sums to one, $P$ is a row-stochastic adjacency matrix.

**The graph Laplacian.**  Given $P$, the random-walk normalized graph Laplacian is

$$\mathcal{L} = I - P,$$

where $I \in \mathbb{R}^{T \times T}$ is the identity matrix. Since $P$ is row-stochastic, applying $\mathcal{L}$ to a signal $x \in \mathbb{R}^T$ measures its deviation from the local row-normalized average over neighbors.

**Diffusion as the discrete heat equation.**  Diffusion on a graph evolves $x(t)$ according to

$$\frac{d}{dt}x(t) = -\mathcal{L}x(t),$$

whose solution progressively smooths $x$ along the edges. A single explicit Euler step with step size $\Delta t = 1$ gives

$$x_{t+1} = x_t - \mathcal{L}x_t = x_t - (I - P)x_t.$$

**Connecting diffusion to our mechanism.**  This update matches exactly what our Laplacian head together with the residual connection in equation 2 performs:

$$X' = X - \mathcal{L}(V) = X - (I - P)V.$$

The residual connection in equation 2 is written with the opposite sign convention, i.e. $X' = X + \mathcal{L}(V)$. This is purely a notational choice: the sign can always be absorbed into the learnable matrices (for example, replacing $W_V$ by $-W_V$ leaves the mechanism mathematically equivalent).

**Consequences for token geometry.**  Repeated application of the normalized graph Laplacian derived from attention scores drives token representations toward their locally averaged state, reducing within-sequence variance and moving them closer to the collapses described in Subsection 3.

## 6 RELATED WORKS

**Neural Collapse**  NC is a set of phenomena that describe the training dynamics of deep classifier networks in the terminal phase (Papyan et al., 2020). It reveals the emergence of a distinctive

geometry in the final-layer representations. NC has been widely studied in various settings (Han et al., 2022; Zhu et al., 2021; Zhou et al., 2022a;b; Rangamani et al., 2023; Jacot et al., 2024; Wu & Papyan, 2024; Fisher et al., 2024; Yan et al., 2024). Recent work by Súkeník et al. (2025) shows that NC is optimal for transformers and ResNet, and several works focus on inducing NC-related properties in the last-layer features and weights (Markou et al., 2024; Chen et al., 2024). However, these works do not consider the collapse of tokens within the same sequence. Many other works leverage the properties of NC to perform specific tasks (Ammar et al., 2024; Pham et al., 2025).

**Rank Collapse**   Our work is closely related to the widely studied phenomenon of *rank collapse*, where the dimension of token embeddings progressively decreases as they pass through transformer blocks (Dong et al., 2023; Noci et al., 2022; Saada et al., 2025; Geshkovski et al., 2025; Kirsanov et al., 2025; Zhou et al., 2025; Bruno et al., 2025). Rank collapse is commonly regarded as a degenerate behaviour to be mitigated because it causes training difficulties (Noci et al., 2022) and limits the model's expressivity (Dong et al., 2023; Barbero et al., 2024). Our work shows that tokens could collapse into a specific geometry that is beneficial for classification, so that collapse is not universally bad. Relatedly, Geshkovski et al. (2024) prove that self-attention collapses tokens into various geometries depending on the initial conditions. However, their work does not investigate how those geometries impact performance. Viswanathan et al. (2025) take one step further to measure the geometric properties of token embeddings in large language models. They show that input prompts with a higher loss produce token embeddings with higher intrinsic dimension. This finding agrees with our hypothesis that NTC0 benefits classification.

**Token Collapse in Autoregressive Language Models**   A recent work by Zhang et al. (2025) observes a low-rank structure in the token embeddings of transformers trained for next-token prediction. More explicitly, Zhao et al. (2025) show that next-token prediction implicitly favors learning logits with a sparse-plus–low-rank structure, where the low-rank component becomes dominant during training and depends only on the support pattern of the context–token co-occurrence matrix. Consequently, when projected onto an appropriate subspace, contexts that share similar next-token supports collapse toward shared low-dimensional directions—a phenomenon they term *subspace collapse*. Our results in Section 4.6.2 empirically support their theories: the Laplacian heads could induce subspace collapse more efficiently, leading to improved downstream performance.

**Oversmoothing in Graph Neural Networks**   Since transformers can be viewed as graph neural networks (GNNs) (Joshi, 2025), our work is related to the oversmoothing phenomenon in GNNs (Roth & Liebig, 2024; Rusch et al., 2023; Álvaro Arroyo et al., 2025). Like rank collapse, oversmoothing is regarded as an issue to be mitigated (Roth et al., 2024; Rusch et al., 2022; Nguyen et al., 2023; Bodnar et al., 2023; Wang et al., 2022). In particular, Rusch et al. (2022) addresses oversmoothing by modifying the underlying GNN dynamics. From this perspective, our modification can be understood as discrete heat diffusion on graphs, a connection that we discuss in Section 5.

## 7   CONCLUSION

Neural Token Collapse represents an ideal state where token embeddings collapse to a simplex-ETF geometry that maximizes class separability. Standard transformers diverge from this geometry because attention moves tokens by inflating their means rather than directly along their variance directions. To address this, we proposed a simple Laplacian-based modification of attention that enables tokens to move along their variance directions, promoting a more efficient convergence toward the desired structure. The Laplacian head can be interpreted as performing diffusion—via a heat equation—of token embeddings over the graph implicitly learned by the attention mechanism. Experiments across datasets show that this modification moves token embeddings markedly closer to NTC and yields consistent gains in performance. A promising direction for future work is to explore how these mechanisms can be modified further to fully realize the envisioned NTC ideal.

It is important to note that while NTC is ideal for classification, the optimal representation geometry for other tasks might be different. By allowing tokens to move more freely along their variance directions, the Laplacian mechanism can, in principle, encourage tokens to deviate from their means. Therefore, another important direction for future work is to characterize the optimal geometries for other tasks and study whether the Laplacian can steer representations towards those ideals.

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

# A    PCA Visualization Method

---

**Algorithm 1** PCA Projection of Tokens to $\mathbb{R}^2$

---

**Require:** $X \in \mathbb{R}^{B \times T \times d}$
1: $X \leftarrow \text{reshape}(X, [B \cdot T, d])$
2: $\mu \leftarrow \text{mean}(X, \text{axis} = 0)$
3: $U, \Sigma, V^\top \leftarrow \text{SVD}(X - \mu)$
4: $V_2 \leftarrow V[:, 0:2]$
5: **return** $(X - \mu) \cdot V_2$

---

# B    Experiment Details

## B.1    Image Classification

To produce results in Table 1, we trained the ViT-B model from Touvron et al. (2022) The model consists of 12 blocks, with 12 attention heads in each block, and an embedding dimension of 768. The number of trainable parameters is around 86.6 million. For each dataset, we sweep the peak learning rate over the set $\{4e-5, 3e-4, 5e-4, 3e-3, 4e-3\}$ and weight decay over the set $\{0.01, 0.02, 0.05\}$ and select whichever combination that works the best. Many other hyperparameters (such as the mixup (Zhang et al., 2018) $\alpha$) were selected following the training recipe detailed in Touvron et al. (2022). Depending on the dataset, we use RandAugment (Cubuk et al., 2019) or 3-Aug Touvron et al. (2022) for data augmentation and AdamW (Loshchilov & Hutter, 2019) or LAMB (You et al., 2020) as the optimizer. All models in Table 1 were trained for 300 epochs using 3 random seeds. Full details of our training setup, including the hyperparameters that were eventually selected for the experiments, are provided in Table 3.

|  | **CIFAR-10** | **CIFAR-100** | **ImageNet** |
|---|---|---|---|
| Loss | Cross Entropy | Cross Entropy | Binary Cross Entropy |
| Optimizer | AdamW | AdamW | LAMB |
| AdamW $\beta_1$ | 0.9 | 0.9 | 0.9 |
| AdamW $\beta_2$ | 0.99 | 0.99 | 0.999 |
| Starting Learning Rate | 3e-6 | 3e-6 | 1e-3 |
| Peak Learning Rate | 3e-4 | 3e-4 | 3e-3 |
| Minimum Learning Rate | 0 | 0 | 1e-6 |
| Weight Decay | 0.02 | 0.02 | 0.02 |
| Drop Path Rate | 0.1 | 0.1 | 0.3 |
| Batch Size | 512 | 512 | 2048 |
| Gradient Clipping | 1.0 | 1.0 | 1.0 |
| LR Scheduler | Cosine Annealing | Cosine Annealing | Cosine Annealing |
| Warmup Epochs | 5 | 5 | 5 |
| Data Augmentation | RandAugment | RandAugment | 3-Aug |
| Mixup $\alpha$ | 0.8 | 0.8 | 0.8 |
| Mixup Probability | 1.0 | 1.0 | 1.0 |
| Input Size | 32×32 | 32×32 | 224×224 |
| Patch Size | 4×4 | 4×4 | 16×16 |
| Precision | float32 | float32 | bfloat16 |

Table 3: Training setup for CIFAR-10, CIFAR-100, and ImageNet with hyper-parameter selection informed by Touvron et al. (2022).

## B.2 Autoregressive Language Modelling

The model architecture we used was based on (Yang et al., 2025; Karpathy, 2025; 2022; Jordan et al., 2024). The model is a decoder-only transformer similar to GPT-2, with the following architectural modification:

- Query-key normalization (Henry et al., 2020; Yang et al., 2025).
- Rotary positional encodings (Su et al., 2023).
- Squared ReLU activation (Nvidia et al., 2024).
- Untied weights for the first-layer embeddings and the last linear layer.

We chose this architecture as the baseline since we found it to have more stable training and converge to a lower validation loss with our limited compute resources. We trained two transformer models with 836M parameters. Each model has Each model was trained on roughly 20B FineWeb-Edu tokens. We followed a WSD learning rate schedule (Hu et al., 2024): the first 5% of the total training duration was used for linear learning rate warmup, and the last 20% of the training duration was used for linear decay. We used the same training setup for both models (Table 4).

| Optimizer | AdamW |
|---|---|
| Peak Learning Rate | 2.5e-4 |
| Minimum Learning Rate | 0.0 |
| Learning Rate Schedule | WSD |
| Batch Size | 0.49M |
| Precision | bfloat16 |

Table 4: Training setup for language modelling on FineWeb-Edu.

After training, we evaluated the models on the following zero-shot tasks: HellaSwag(Zellers et al., 2019), PIQA(Bisk et al., 2019), ARC-Easy(Clark et al., 2018), ARC-Challenge(Clark et al., 2018),OpenBookQA(Mihaylov et al., 2018), WinoGrande(Sakaguchi et al., 2019), RACE(Lai et al., 2017), and SciQ(Johannes Welbl, 2017).

## C   Additional Experiment Results

This section contains additional measurements and visualizations for the models in Section 4.1.

### C.1   PCA Visualizations

For CIFAR100 and ImageNet-1k, we first randomly sample 10 and 3 classes, respectively. Then we sample a batch of 512 images that contain roughly an equal number of images across the selected classes. All PCA visualizations in this section are performed using the output of some layer (usually a block from the later half, or the final normalization layer), as indicated in the plot.

### C.1.1   CIFAR10

**ViT-B (Baseline):**

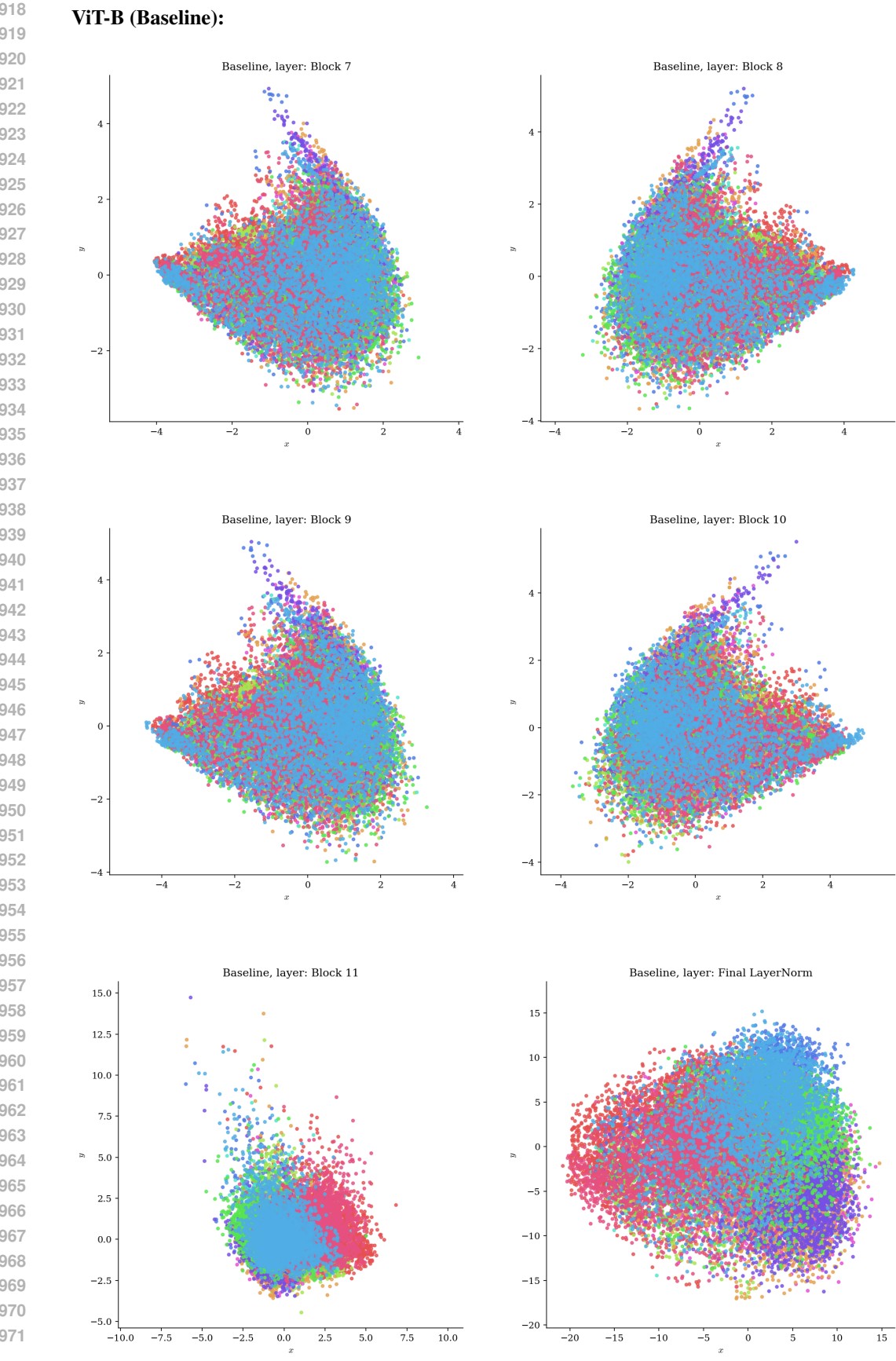

**ViT-1P (Proposed):**

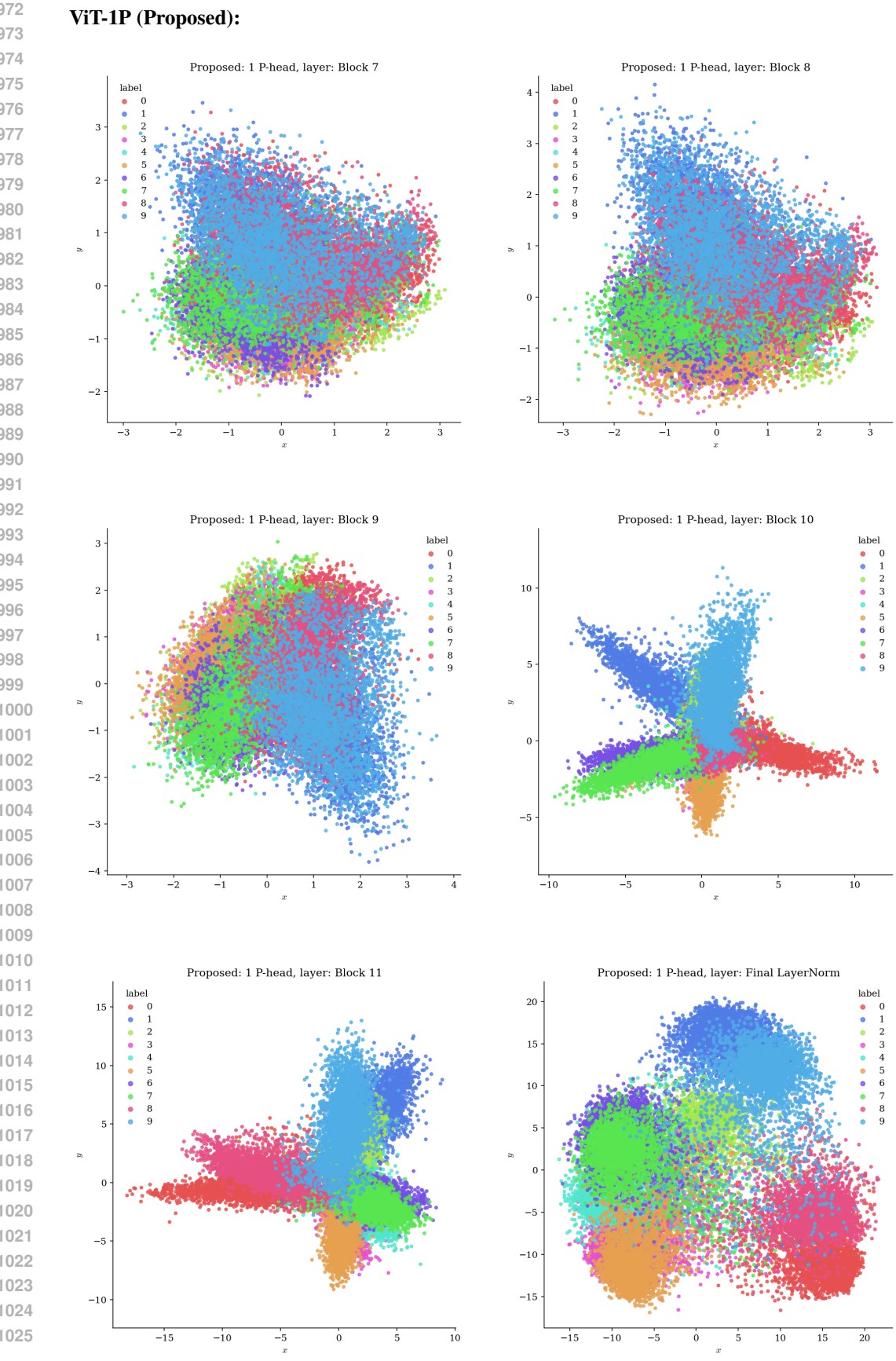

**ViT-3P (Proposed):**

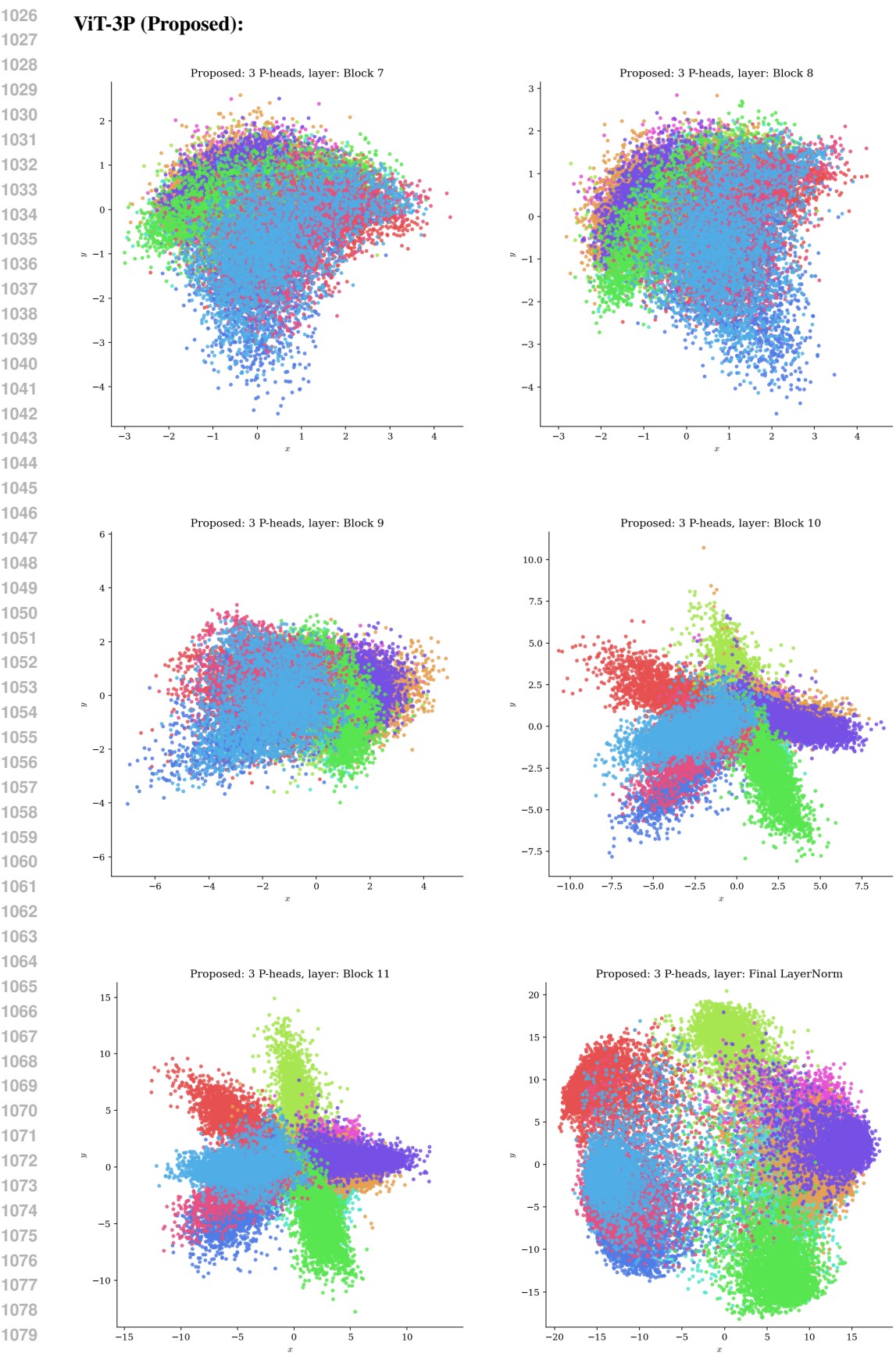

**ViT-mix-depth (Proposed):**

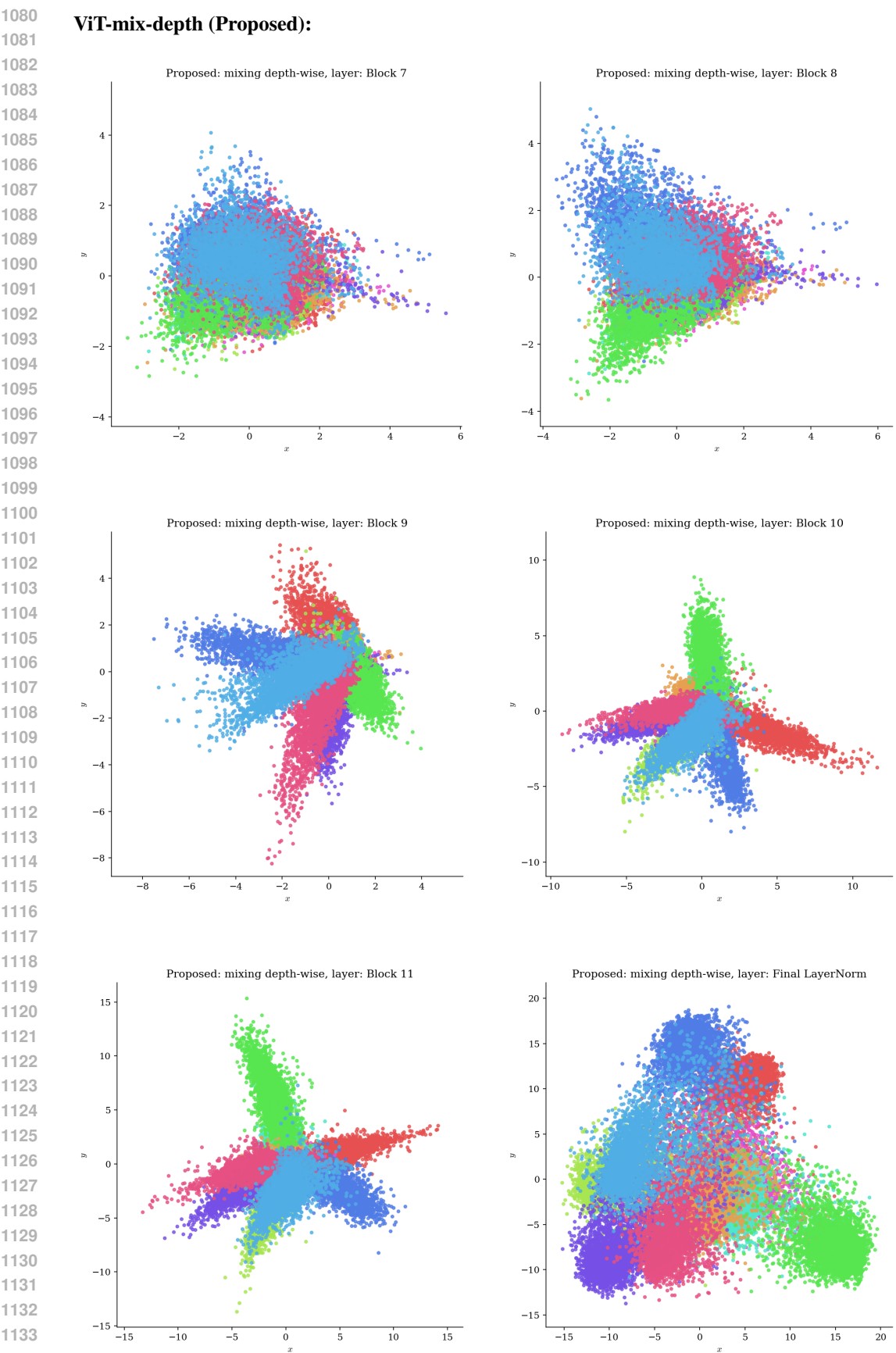

**ViT-0P (Proposed):**

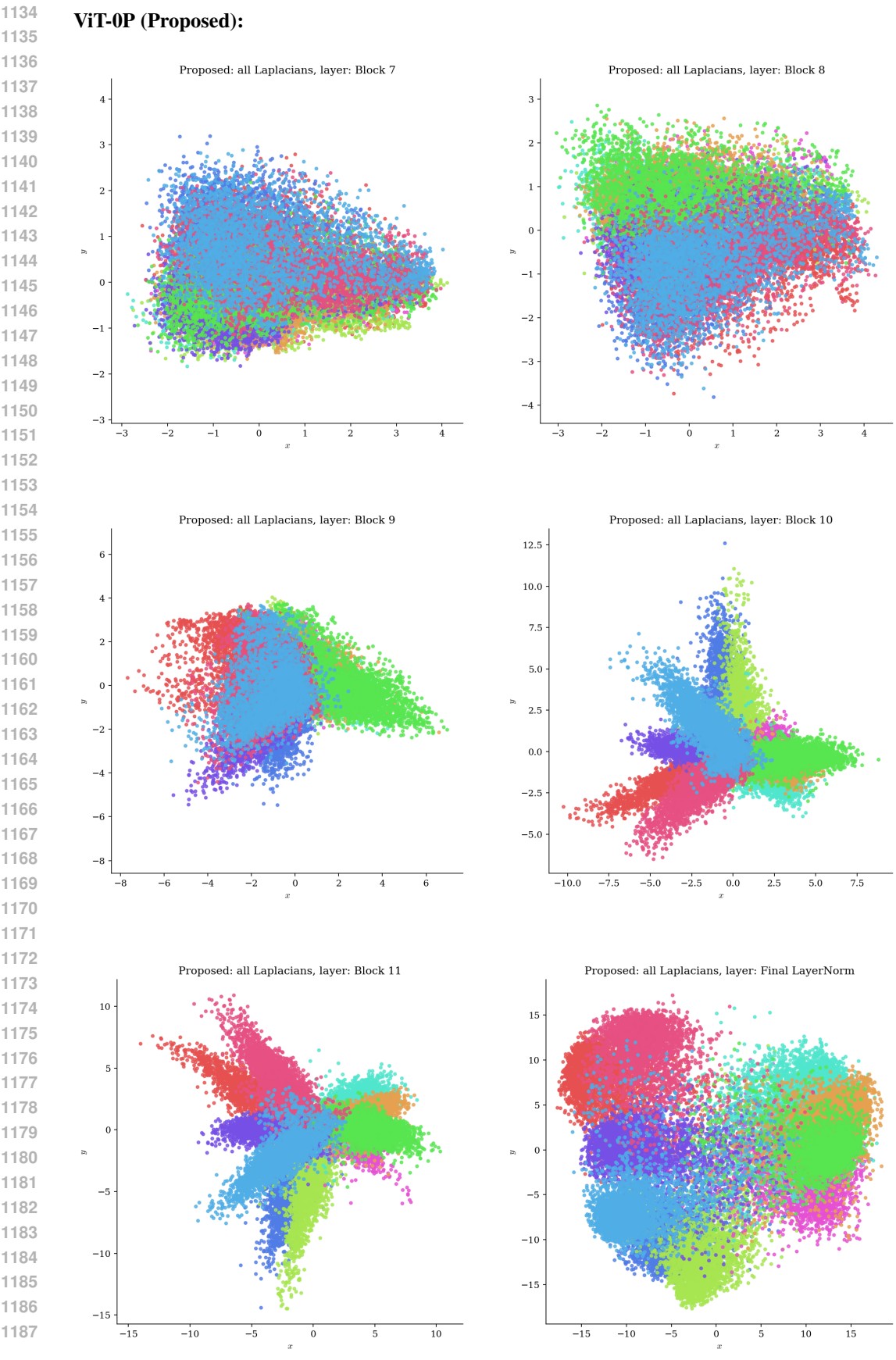

Notice the emergence of the distinct geometry in the token embeddings for the proposed models (block 9 to block 10 for ViT-0P, ViT-1P, and ViT-3P, and block 8 to block 9 for ViT-mix-depth). We do not observe this phenomenon in the baseline model.

### C.1.2 CIFAR100

**ViT-B (Baseline):**

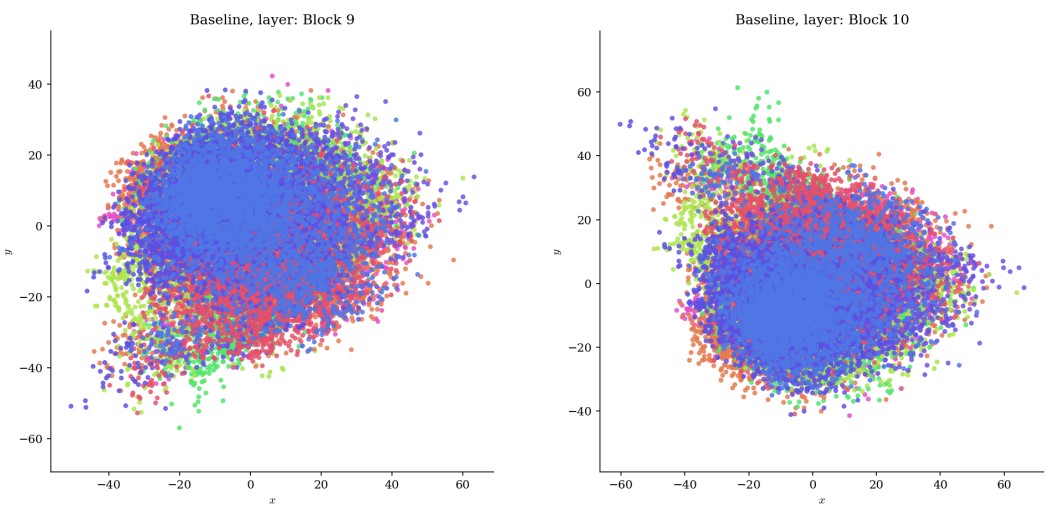

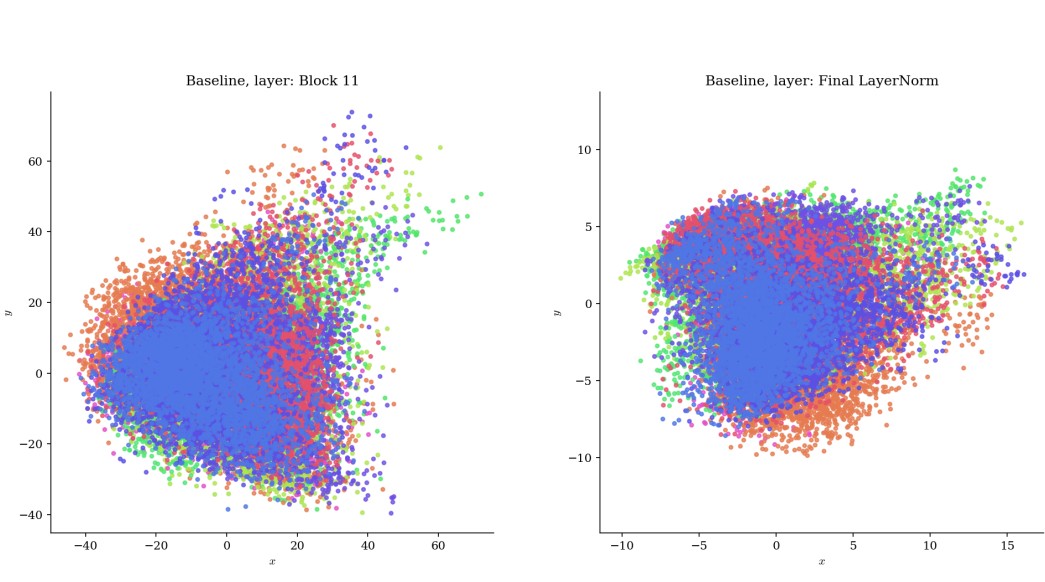

**ViT-0P (Proposed):**

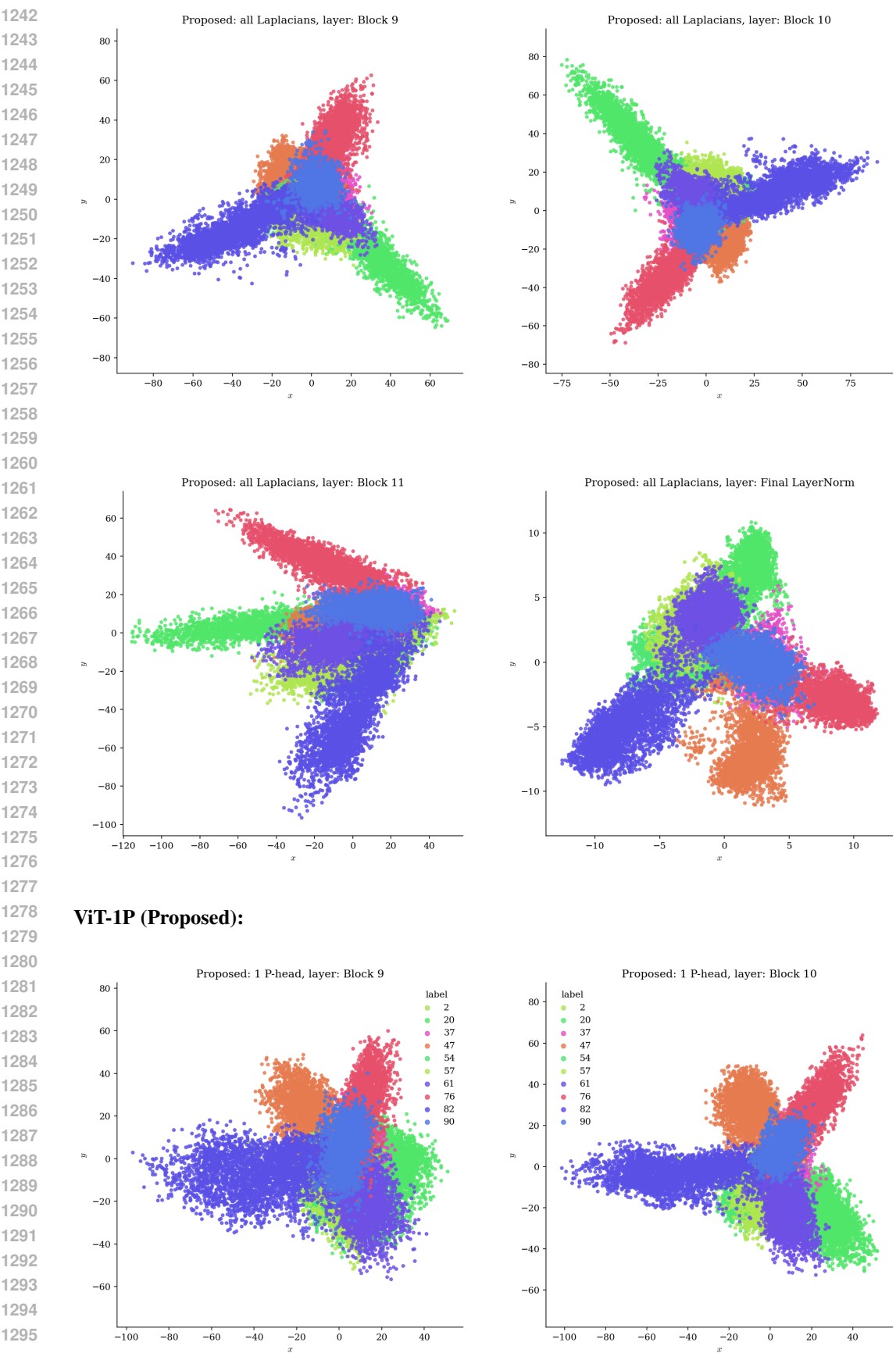

**ViT-1P (Proposed):**

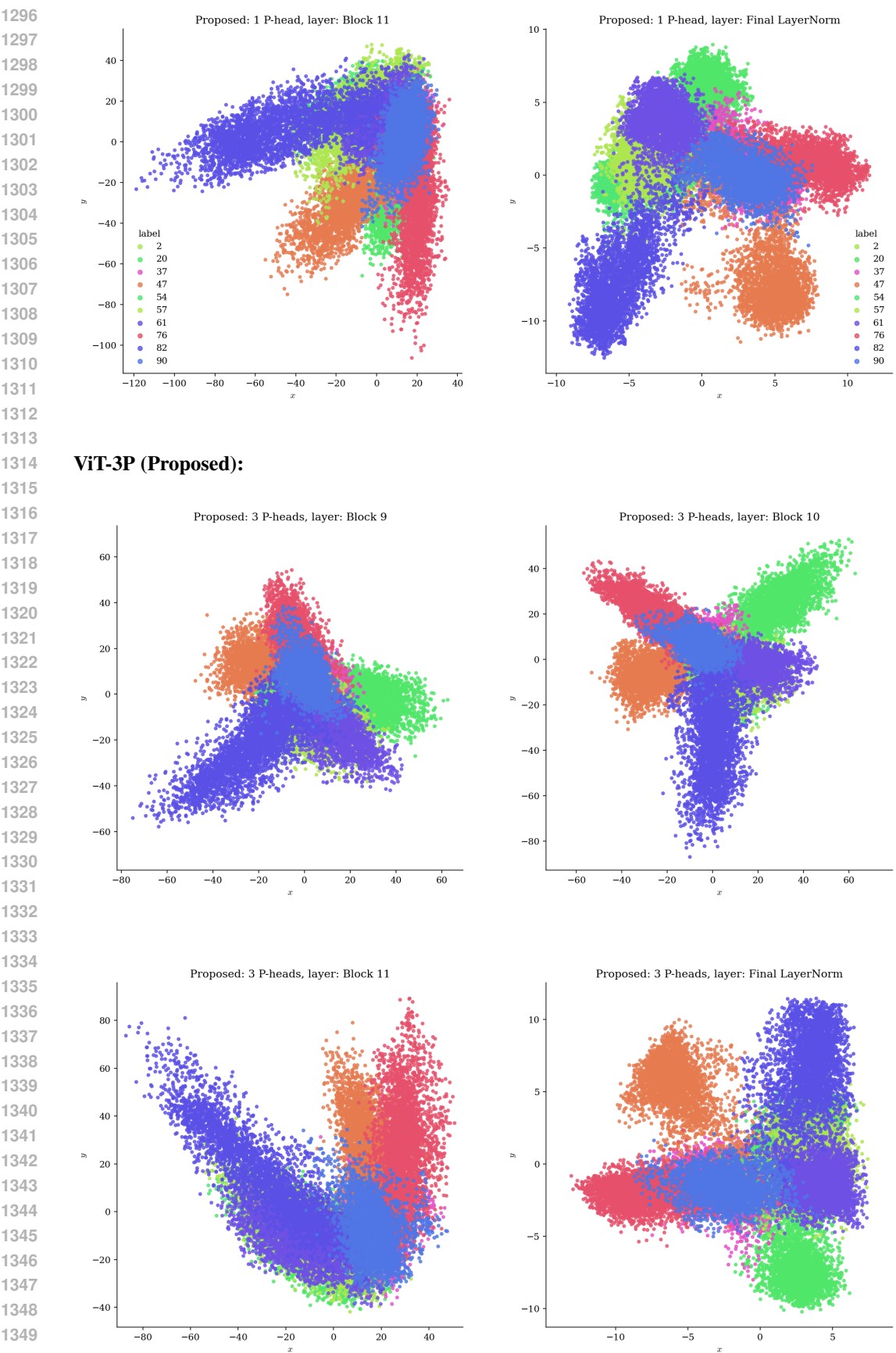

**ViT-3P (Proposed):**

**ViT-mix-depth (Proposed):**

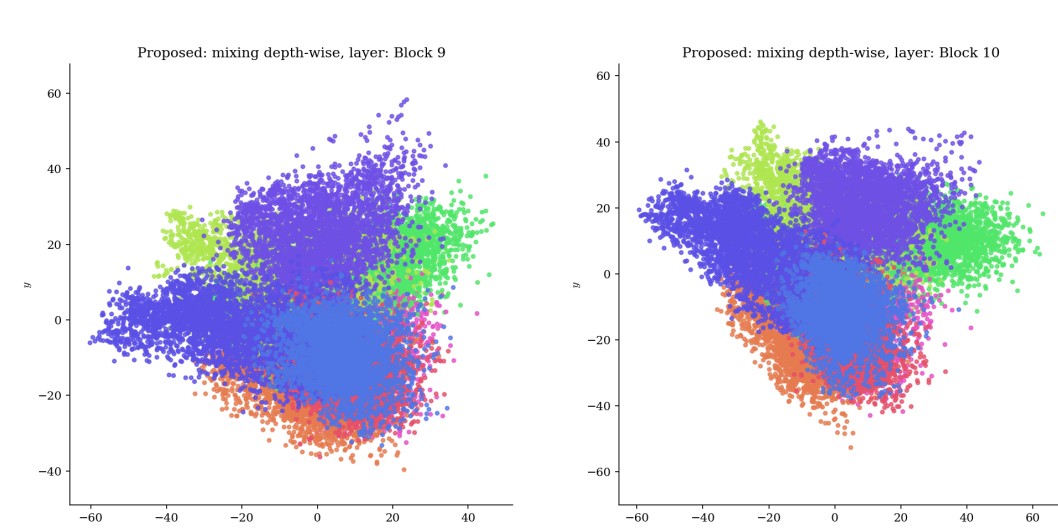

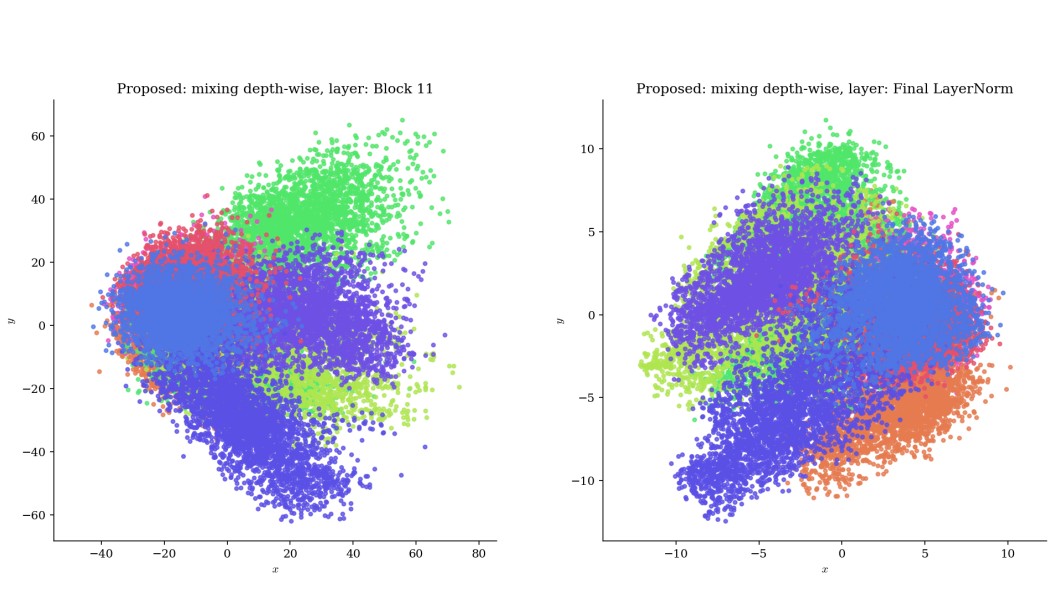

### C.1.3 IMAGENET-1K

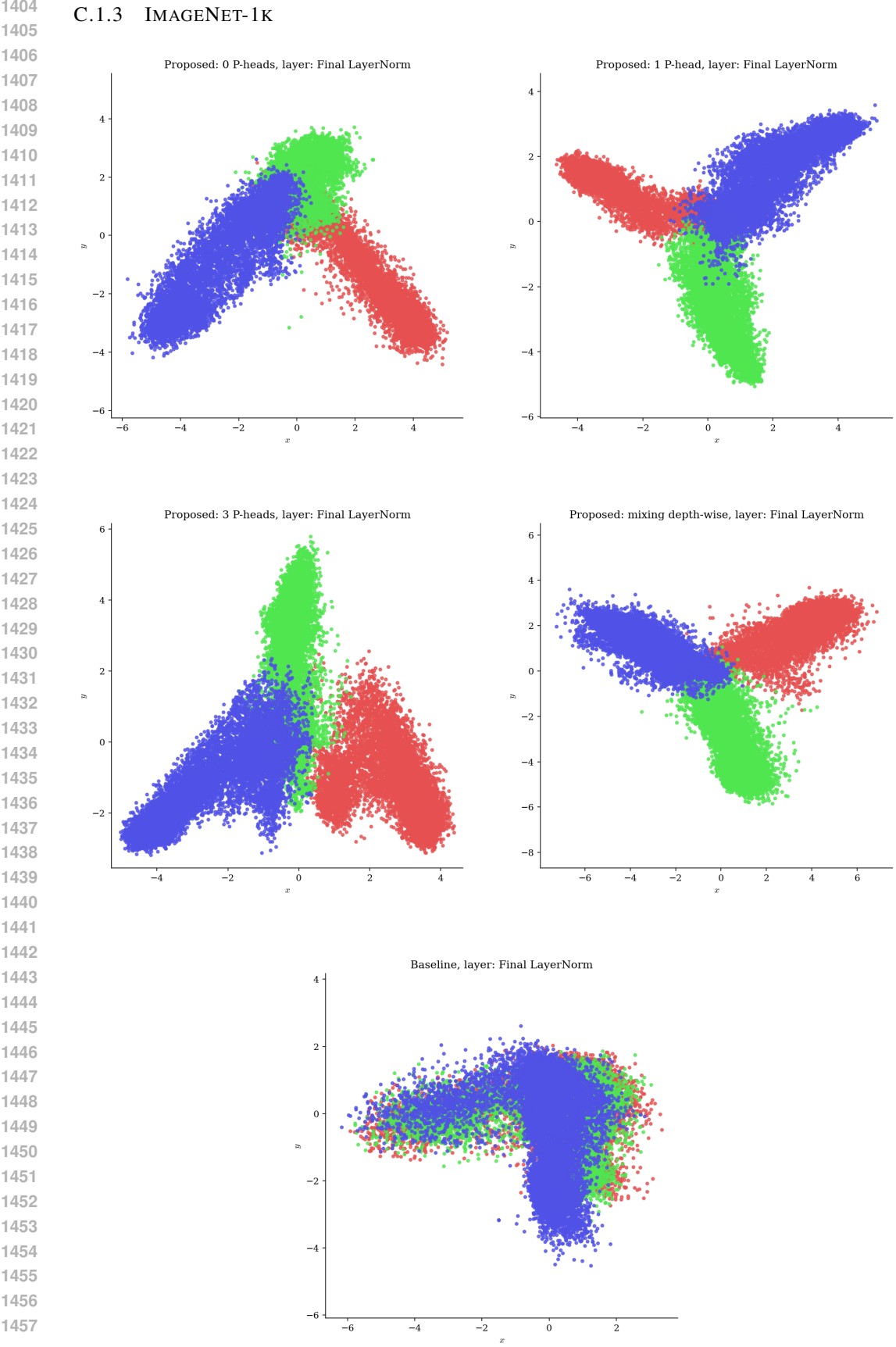

## C.2 ANOVA DECOMPOSITION

### C.2.1 CIFAR10

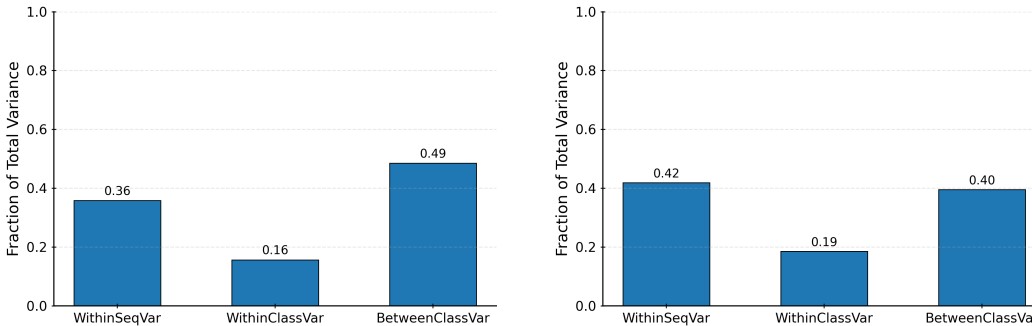

Figure 7: ANOVA decomposition of normalized variance for different model variants on CIFAR-10. Left: ViT-B-3P; Right: ViT-B-Mix-Depth.

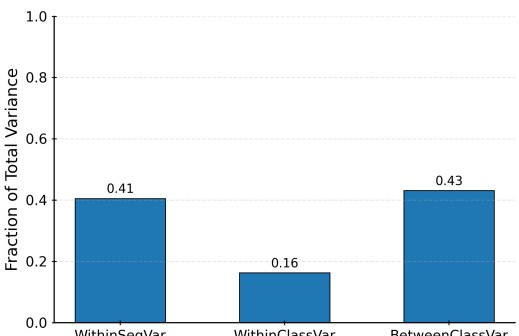

Figure 8: ANOVA decomposition of normalized variance for ViT-B-0P on CIFAR-10.

### C.2.2   CIFAR100

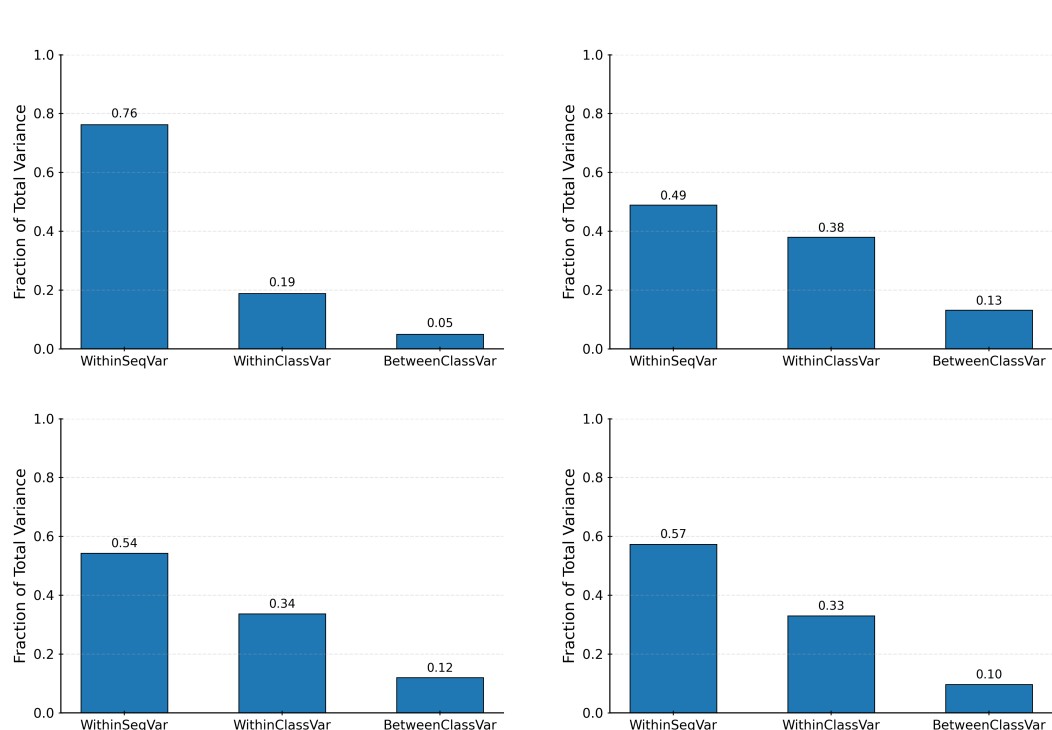

Figure 9: ANOVA comparison of normalized variance for different model variants on CIFAR-100. Top row: ViT-B (left) and ViT-B-1P (right). Bottom row: ViT-B-3P (left) and ViT-B-Mix-Depth (right).

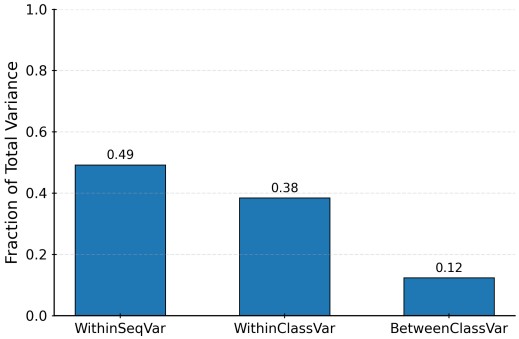

Figure 10: ANOVA decomposition of normalized variance for ViT-B-0P on CIFAR-100.

### C.2.3 IMAGENET-1K

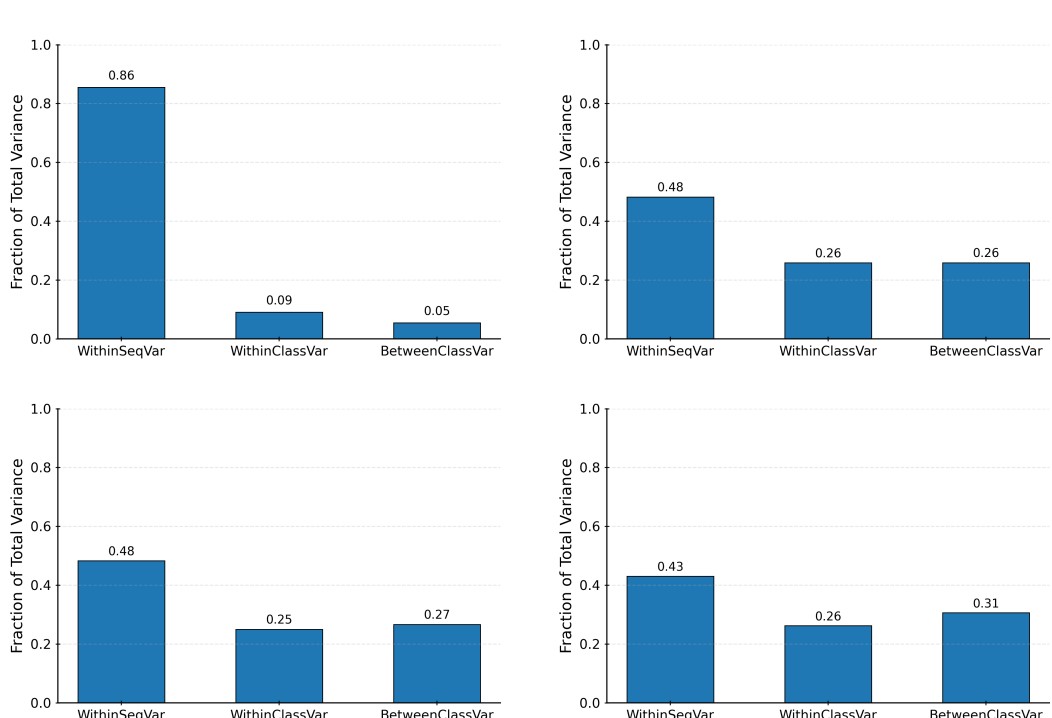

Figure 11: ANOVA comparison of normalized variance for different model variants on ImageNet-1k. Top row: ViT-B (left) and ViT-B-1P (right). Bottom row: ViT-B-3P (left) and ViT-B-Mix-Depth (right).

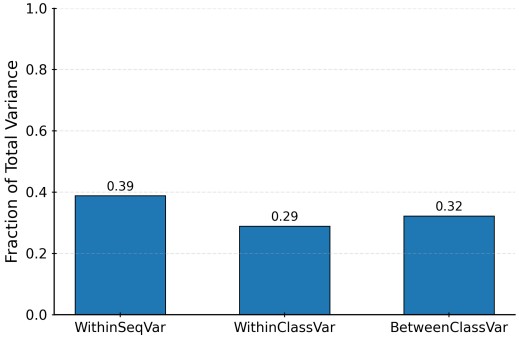

Figure 12: ANOVA decomposition of normalized variance for ViT-B-0P on ImageNet-1k.

## C.3 COSSIM METRIC

### C.3.1 CIFAR10

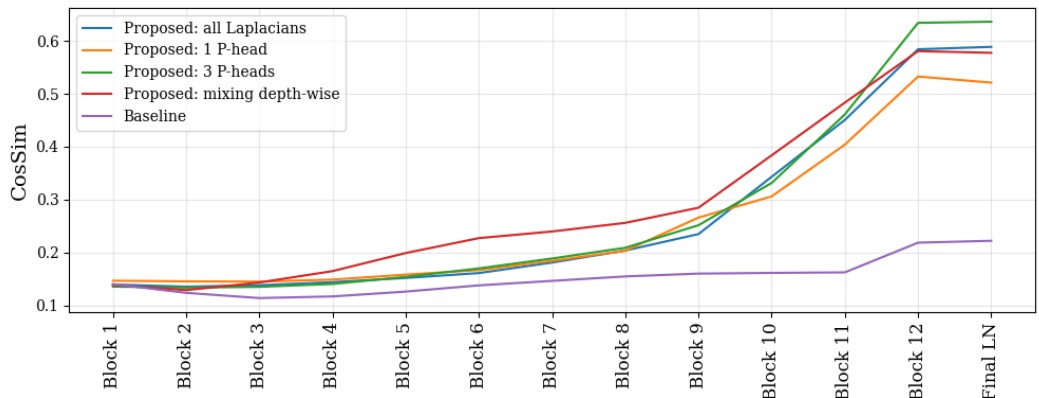

Figure 13: CosSim across depth on CIFAR10.

### C.3.2 CIFAR100

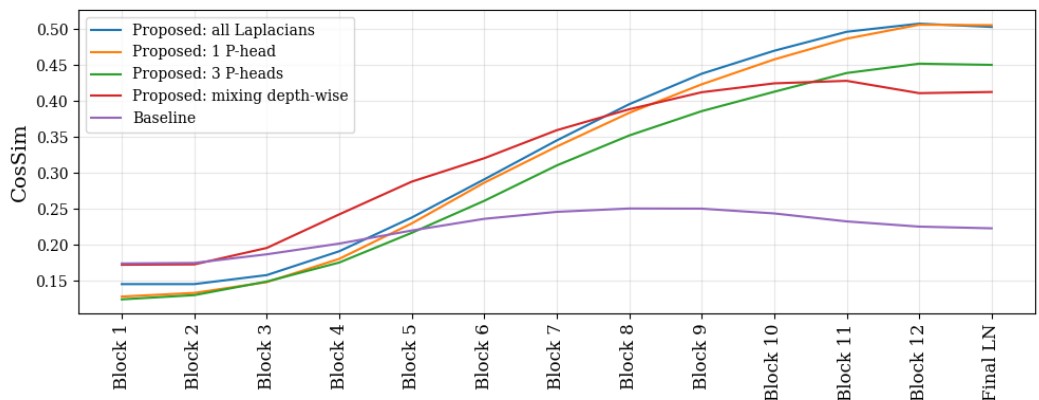

Figure 14: CosSim across depth on CIFAR100.

## C.4 NC METRICS AND VISUALIZATION

### C.4.1 CIFAR100

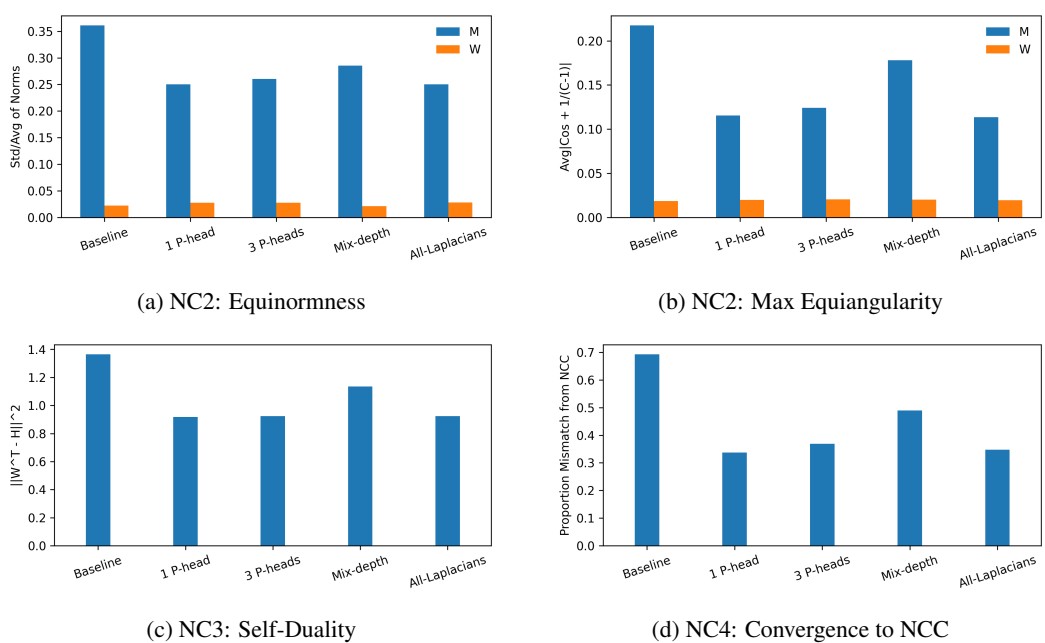

(a) NC2: Equinormness

(b) NC2: Max Equiangularity

(c) NC3: Self-Duality

(d) NC4: Convergence to NCC

Figure 15: Neural Collapse metrics on CIFAR100.

### C.4.2 IMAGENET-1K

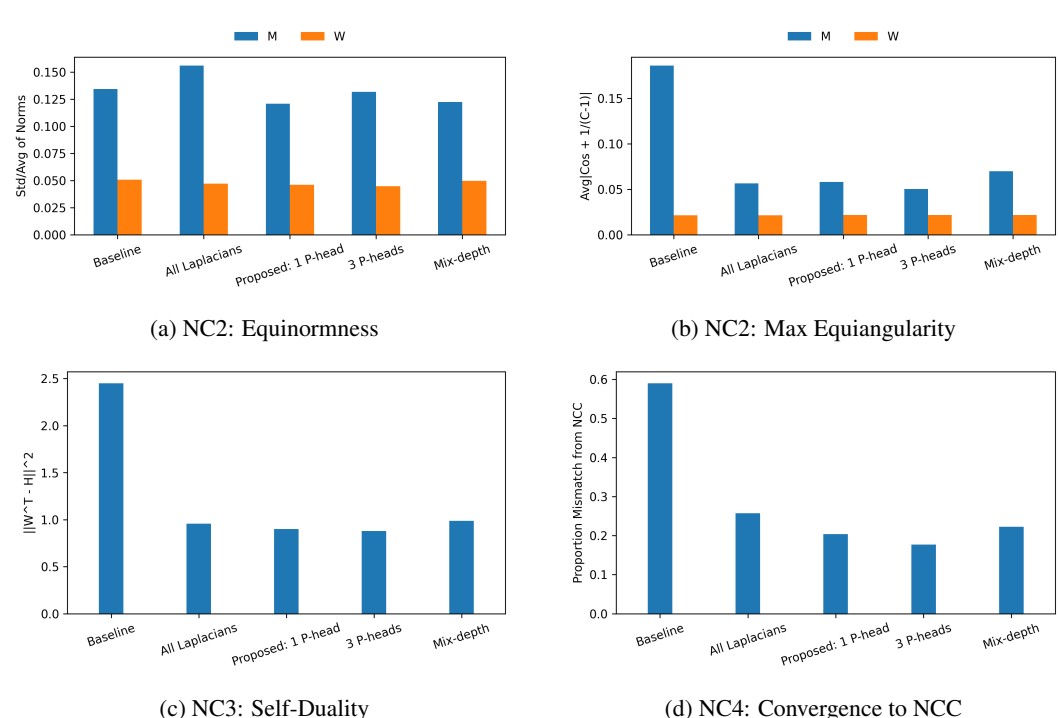

(a) NC2: Equinormness

(b) NC2: Max Equiangularity

(c) NC3: Self-Duality

(d) NC4: Convergence to NCC

Figure 16: Neural Collapse metrics on ImageNet-1k.

## C.5 Visualizations of Projections onto a Simplex

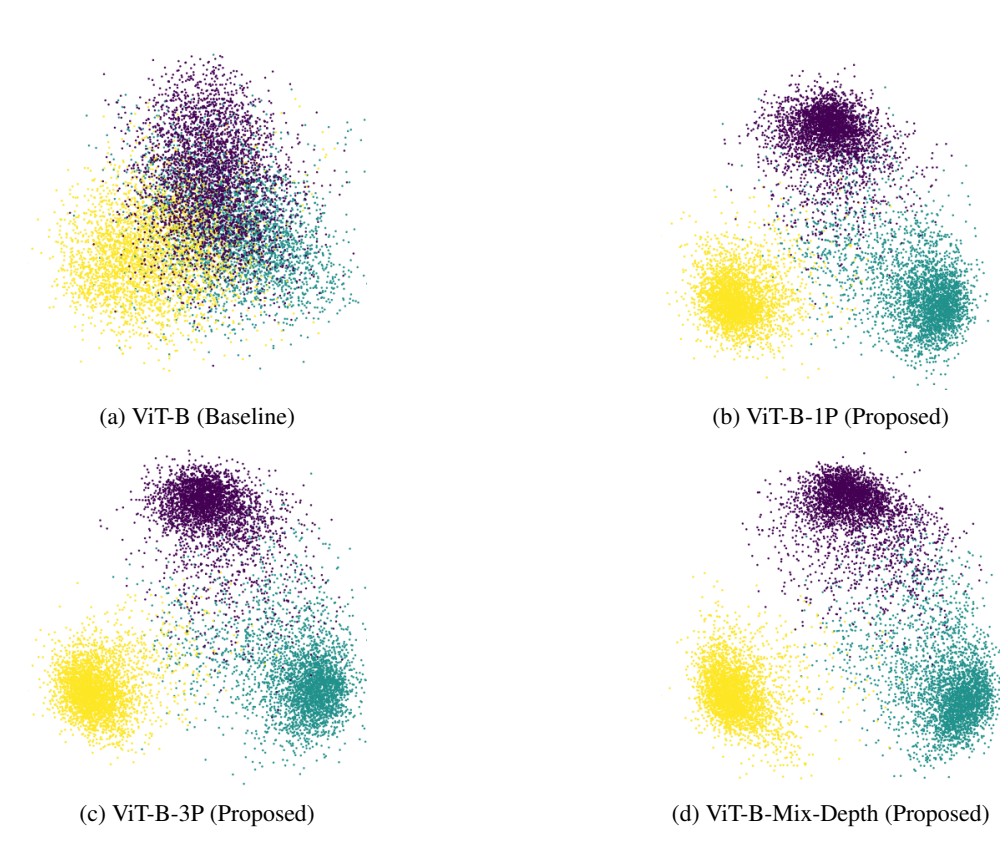

(a) ViT-B-0P (Proposed) on CIFAR-10

(b) ViT-B-0P (Proposed) on CIFAR100

Figure 17: Visualization of projections onto a simplex for ViT-B-0P.

(a) ViT-B (Baseline)

(b) ViT-B-1P (Proposed)

(c) ViT-B-3P (Proposed)

(d) ViT-B-Mix-Depth (Proposed)

Figure 18: Visualization of projections onto a simplex on CIFAR10.

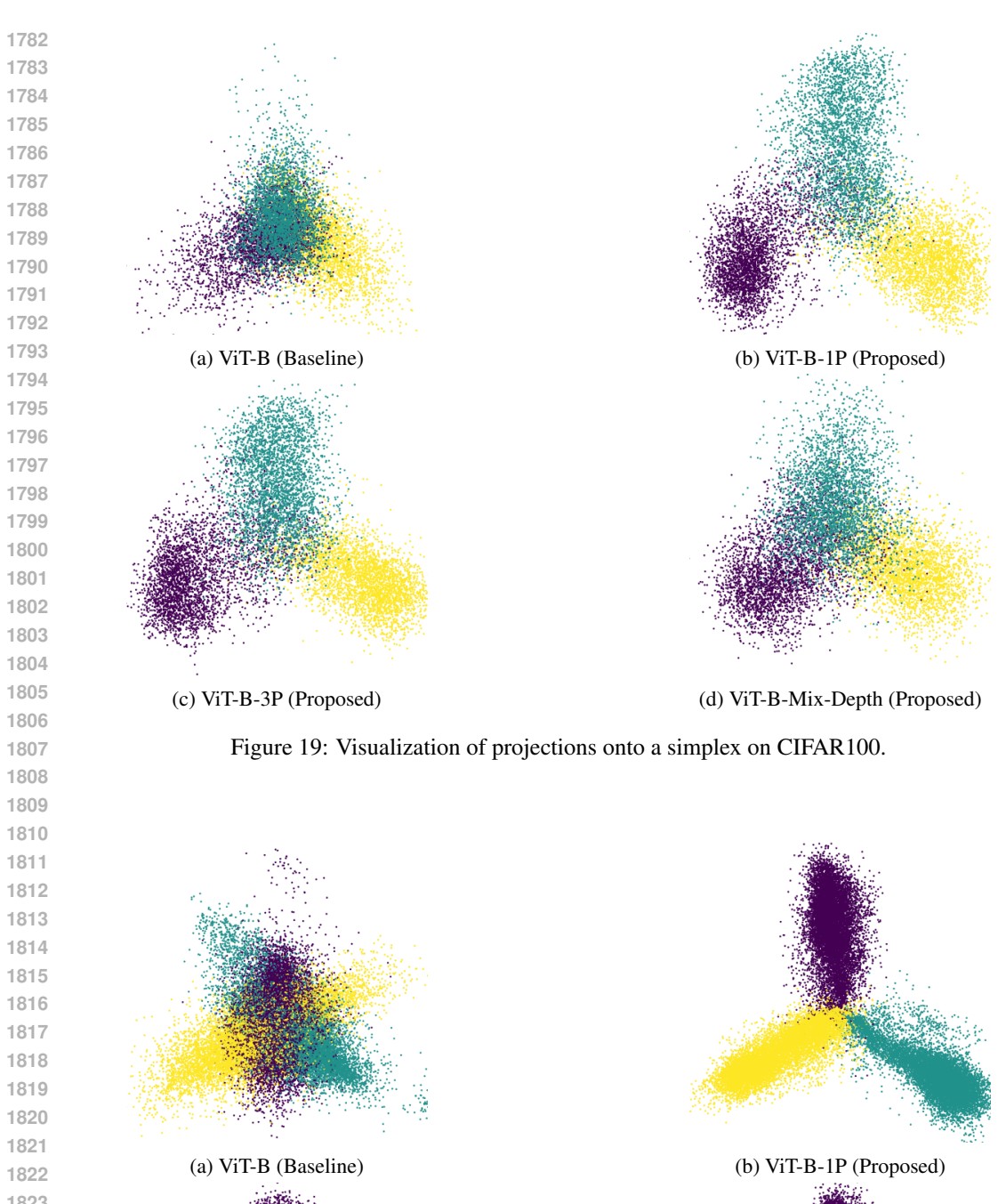

(a) ViT-B (Baseline)

(b) ViT-B-1P (Proposed)

(c) ViT-B-3P (Proposed)

(d) ViT-B-Mix-Depth (Proposed)

Figure 19: Visualization of projections onto a simplex on CIFAR100.

(a) ViT-B (Baseline)

(b) ViT-B-1P (Proposed)

(c) ViT-B-3P (Proposed)

(d) ViT-B-Mix-Depth (Proposed)

Figure 20: Visualization of projections onto a simplex on ImageNet-1k.

## D EMPIRICAL EVIDENCE FOR THE GEOMETRIC ROLE OF THE LAPLACIAN

Given a batch of sequences of token embeddings $X \in \mathbb{R}^{B \times T \times d}$, we define the average signal-to-noise ratio (SNR) of $X$ as

$$\text{SNR}(X) = \frac{1}{B} \sum_{b=1}^{B} \frac{||\text{Mean}(X_b)||_2}{\text{Std}(X_b)},$$

where

$$\text{Mean}(X_b) = \frac{1}{T} \sum_{i=1}^{T} X_{b,i} \quad \text{and} \quad \text{Std}(X_b) = \sqrt{\frac{1}{T} \sum_{i=1}^{T} ||X_{b,i} - \text{Mean}(X_b)||_2^2}$$

Here, $X_b \in \mathbb{R}^{T \times d}$ denote the $b$th sequence in the batch and $X_{b,i} \in \mathbb{R}^d$ denote the $i$th token embedding within the sequence.

The SNR directly measures how large (in $l_2$ norm) the mean of a sequence of tokens is relative to their variance/standard deviation. A larger SNR implies that the sequence is more collapsed since the mean is larger relative to the variance. To validate our geometric interpretation in Sections 2.1 and 2.2 (see Figure 2), we directly measure the SNR of the output of the Layer Normalization module right before the MLP layer. In other words, we measure

$$\text{LayerNorm}\big(X + \mathbf{MHA}(X)\big)$$

for every transformer block. Figure 21 plots the SNR of the ImageNet token embeddings for the baseline and the proposed models as a function of depth. It clearly illustrates that for all models that use the Laplacian heads, the output of the Pre-MLP LayerNorm has significantly higher SNR than the baseline. Moreover, as depth increases, the SNRs for the proposed models grow more drastically. This measurement empirically supports our interpretation of the mechanism by which transformers collapse tokens, and it directly confirms our geometric intuition that the Laplacian induces more efficient collapse of token embeddings.

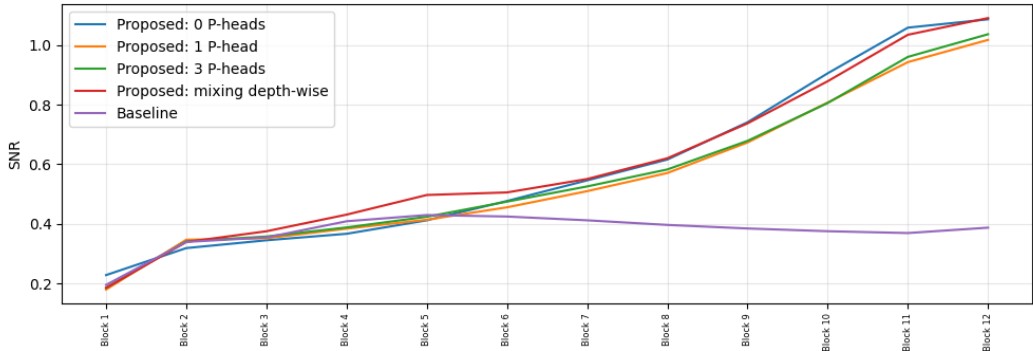

Figure 21: The Laplacian mechanism collapses tokens more effectively.

## E RESULTS FOR ALTERNATIVE WAYS OF MIXING Attn AND $\mathcal{L}$

For a transformer with $n$ blocks and $h$ heads, there are in total $(h + 1)^n$ possibilities of assigning **Attn** and $\mathcal{L}$ to different heads. Although the search space is huge, we hypothesize that the number of options that exhibit meaningful differences is much smaller. Here, we describe a limited subset of options that we experimented with.

The most obvious extension of strategy 1 is to vary the number $m$ of heads that use **Attn** and keep this number the same across depth. The ViT-B model has 12 heads and 12 blocks, and we experimented with $m \in \{0, 1, 3, 6, 9, 12\}$. As discussed in Section 4.6.1 We observed that while $m = 0$ already produces noticeable improvements upon the baseline, incorporating a small number

of standard attention heads sometimes induces further improvements. This agrees with our intuition that it is beneficial to allow movement of tokens in both the mean and variance directions. In our experiments, $m = 1, 3$ consistently produced the best results across all datasets, and as $m$ increased, the model's performance converged to that of the standard transformer (equivalent to $m = 12$). We report the performance for different $m$ values in Table 5.

Table 5: Top–1 test accuracy (%) of models with different numbers of standard attention heads.

| $m$ | CIFAR-10 | CIFAR-100 | ImageNet-1k |
|-----|----------|-----------|-------------|
| 0   | 91.74    | 65.39     | 82.02       |
| 1   | 91.83    | 66.05     | 82.18       |
| 3   | 91.83    | 65.44     | 82.17       |
| 6   | 91.27    | 64.33     | 81.73       |
| 9   | 90.93    | 62.37     | 81.96       |
| 12  | 90.41    | 61.41     | 81.2        |

Another obvious strategy inverts strategy 2 by using $\mathcal{L}$ for all heads in the first half of blocks and use **Attn** for all heads in the second half. However, this strategy consistently produced worse results than strategy 2. We also tried interleaving blocks that only used **Attn** and blocks that only used $\mathcal{L}$. Interestingly, the order in which the two types of blocks is interleaved appeared to impact performance significantly, where the order $\textbf{Attn} \rightarrow \mathcal{L} \rightarrow \textbf{Attn} \rightarrow \mathcal{L}$ consistently performed better. More investigation is needed to understand these phenomena.

# F  NEURAL COLLAPSE

## F.1  METRICS

Let $M \in \mathbb{R}^{d \times C}$ be the matrix whose columns are the class means $\{\mu_i : 1 \leq i \leq C\}$ and $W \in \mathbb{R}^{C \times d}$ be the weight matrix of the final-layer classifier. We quantify NC2 - NC4 following Han et al. (2022):

- **NC2 (Equinorm and Maximal Equiangularity)**:
  - *Equinorm*: Measures how uniform the vector norms are within the class means or weights, using the coefficient of variation (CoV):
  $$\frac{\text{std}(\|\mu_c\|)}{\text{mean}(\|\mu_c\|)} \quad \text{and} \quad \frac{\text{std}(\|w_c\|)}{\text{mean}(\|w_c\|)},$$
  where $w_c$ is the classifier weight vector corresponding to class $c$.
  - *Maximal Equiangularity*: Measures how close the vectors are to forming a maximally equiangular tight frame (ETF):
  $$\frac{1}{C(C-1)} \sum_{i \neq j} \left| \langle \hat{v}_i, \hat{v}_j \rangle + \frac{1}{C-1} \right|,$$
  where $\hat{v}_i$ and $\hat{v}_j$ are $\ell_2$-normalized class means or weight vectors. A lower value indicates greater conformity to an ETF structure.
- **NC3 (Self-Duality)**: Measures the alignment between the classifier weights and the centered class means:
  $$\left\| \frac{W^T}{\|W^T\|_F} - \frac{M'}{\|M'\|_F} \right\|_F^2,$$
  where $M' = M - \mu_G \mathbf{1}^T$ is the matrix of class means centered by their global mean $\mu_G$.
- **NC4 (Convergence to NCC)**: Measures how close the learned classifier is to a Nearest Class Center (NCC) classifier:
  $$1 - \frac{1}{N} \sum_{i=1}^{N} \mathbb{1} \left[ \arg\max f(x_i) = \arg\min_c \|h_i - \mu_c\| \right],$$
  where $f(x_i)$ are the logits, $h_i$ is the feature of sample $x_i$, and $\mu_c$ is the mean feature for class $c$.

## F.2 VISUALIZATION OF PROJECTION ONTO SIMPLEX ETF

Each token embedding is first projected onto the classifier $W$ for a random subset of three classes, then the result is projected again onto a two-dimensional representation of a three-dimensional simplex ETF. The result is visualized with each point colored according to its ground truth class. This visualization aims to illustrate the conformity of token embeddings to a simplex ETF.

---

**Algorithm 2** Projection of Tokens to a simplex ETF

---

**Require:** $X \in \mathbb{R}^{B \times T \times d}, W \in \mathbb{R}^{C \times d}$
1: $X \leftarrow \text{reshape}(X, [B \cdot T, d]), W' \in \mathbb{R}^{3 \times d} \leftarrow \text{random sample}(W)$
2: $U, S, V^T = \text{SVD}(\text{normalize}(W'))$
3: $A \leftarrow \sqrt{2} \cdot \begin{bmatrix} \frac{1}{2} & -\frac{1}{2} & 0 \\ 0 & 0 & \frac{\sqrt{3}}{2} \end{bmatrix} \cdot \left(I_3 - \frac{1}{3}\mathbf{1}\mathbf{1}^\top\right)$
4: output $AUV^T X^T$

---

