# OpenReview forum: "A Two-Character Change in Transformer Architecture Promotes Ideal Token Geometry"
_ICLR.cc/2026/Conference — Submitted to ICLR 2026_

### Official Review · Reviewer_XAcD · 2025-10-30

**Soundness:** 3
**Presentation:** 2
**Contribution:** 3
**Rating:** 6
**Confidence:** 3

**Summary:**

The paper proposes a Laplacian-based modification to the transformer attention mechanism that encourages token embeddings to evolve toward a beneficial geometric structure called Neural Token Collapse (NTC). By interpreting attention as diffusion on a graph, the method introduces a Laplacian “head” that smooths token representations along meaningful variance directions rather than averaging them, improving both representation structure and classification accuracy. Experiments on CIFAR-10, CIFAR-100, and ImageNet-1k show consistent performance gains over standard Vision Transformers. The authors connect their approach to related concepts such as neural collapse, rank collapse, and oversmoothing in graph neural networks, suggesting that controlled token collapse can enhance model efficiency and separability rather than hinder it.

**Strengths:**

- Originality: There are a few points of novelty in this paper: exploiting token collapse as an improvement of the transformer architecture, rather than an issue, is surely a clean idea that I have not seen before. Along this line, the physical interpretation as graph Laplacian diffusion makes it more grounded to known concepts.
- Clarity: The problem is stated clearly in the introduction and raised questions and goals are well referenced when addressed along the whole text.
- Quality: This paper proposes an hypothesis grounded on the understanding of how transformers works, implements a modification to the architecture that verify the hypothesis and, importantly, as a result improve the performance of the model. This is how a paper on interpretating neural networks should be structured.
- Significance: Principled improvements to transformers architectures are highly relevant in a moment where available human training datasets are being saturated.

**Weaknesses:**

1. The discussion in 2.1 and 2.2 has been heavily motivated by Figure 2. While the figure neatly conveys the idea of how a single attention block modifies the token’s relative positions, it is not clear to me that this would work as neatly in high-dimensional spaces. Some empirical support or citations for the statements made in 2.1 and 2.2 can improve the presentation.
2. The experiment part seems a little bit underdeveloped: the authors consider only one vision model and three datasets, which are used both for training and for showing results. It would also seem natural to discuss what would be the result of applying this method to language models/next token prediction tasks.
3. Presentation-wise, the authors might make a slight effort in improving how they present their experimental results (see questions below for more detail).

**Questions:**

1. While the authors give some details about training, since the method involves a different paradigm, some comparison on the training metrics with the base model (like just the number of epochs) could be useful?
2. The mixing of attention and laplacian seems a bit arbitrary. Appendix D seems to discuss more mixing trials, but I wonder if there is a way of framing this in a more principled way? Would this mixing have to be adapted at each task?
3. Some discussion on extending this framework to next token prediction/autoregressive tasks can be helpful for a more impactful paper. Do the authors think such an extension is possible, i.e., is NTC ideal for next token prediction? Some empirical validation by training small transformers using Laplacian heads would be great.
4. Some figures might benefit a little bit more cosmetics:
- Figure 3 has two diagrams with identical labels and no reference to top/bottom in neither text nor caption
- Figure 4 some labels are cut.
- Figure 5 the legend for “M” and “W” is explained in the caption but might be made more explicit in the plot. Y labels are not in latex despite using equations.
5. In the classification performance part, the datasets used are the same ones used in training. Would it be possible to see this on an unseen dataset?
6. This paper https://arxiv.org/pdf/2408.15417 seems relevant and I haven’t seen it in the relevant works sections.

---

> ### Author Response · Authors · 2025-11-26
>
> We would like to extend our gratitude to the reviewer for highlighting the strengths of our paper, providing valuable feedback, and raising important questions. Below, we address each of the reviewer's point through newly added materials or discussions:
>
> &nbsp;
>
> > ### This paper https://arxiv.org/pdf/2408.15417 seems relevant and I haven’t seen it in the relevant works sections.
>
> &nbsp;
>
> We apologize for our omission. We have included a discussion (attached below) in the Related Works section of this insightful and obviously relevant work.
>
> &nbsp;
>
> ### Token Collapse in Autoregressive Language Models
>
> A recent work [1] observes a low-rank structure in the token embeddings of transformers trained for next-token prediction. More explicitly, [2] show that next-token prediction implicitly favors learning logits with a sparse-plus–low-rank structure, where the low-rank component becomes dominant during training and depends only on the support pattern of the context–token co-occurrence matrix. Consequently, when projected onto an appropriate subspace, contexts that share similar next-token supports collapse toward shared low-dimensional directions—a phenomenon they term subspace collapse. Our results in Section 4.6.2 empirically support their theories: the Laplacian heads could induce subspace collapse more efficiently, leading to improved downstream performance.
>
>
> &nbsp;
>
> > ### The experiment part seems a little bit underdeveloped: the authors consider only one vision model and three datasets, which are used both for training and for showing results. It would also seem natural to discuss what would be the result of applying this method to language models/next token prediction tasks.
>
> &nbsp;
>
> We acknowledge the reviewer’s concern. In response, we have added results for an almost-1B parameter transformer language model trained for next-token prediction on 20B FineWeb-Edu tokens. Our new results (Table 2 in paper, attached below) show that after training on the same number of tokens, the proposed modification consistently achieves better performance than the baseline on a variety of zero-shot downstream tasks.
>
> **Table: Zero-shot results of 836M GPT2-style decoder-only transformers on downstream datasets.**
> *Metric = accuracy (higher is better).*
>
> | Model                         | ARC-Easy | ARC-Challenge | HellaSwag | PIQA  | RACE | OpenBookQA | WinoGrande | SciQ |
> |------------------------------|----------|----------------|-----------|-------|------|-------------|-------------|------|
> | Baseline (836M)              | 63.8     | 31.48          | 36.22     | 68.55 | 30.62 | 33.2       | 53.99       | 84.1 |
> | **Proposed: 5 P-heads (836M)** | **64.56** | **32.08**      | **36.4**  | **68.71** | **32.15** | **34.6** | **54.78** | **85.2** |
>
>
> &nbsp;
>
> In addition, we have updated Table 1 with new ImageNet-1k results (attached below) that show greater improvements upon the baseline. Originally, our experiments showed an improvement of 0.3% to 0.6% in top-1 validation accuracy upon a baseline accuracy of 80.32%. The new results show an improvement of **0.8% to almost 1% upon a higher baseline of 81.2%**. These new results are obtained by further optimizing the hyper-parameters and training recipe for the baseline, which we have detailed in Appendix B.
>
> **Table: Top-1 ImageNet-1k accuracy (%) of models.**
>
> | Model | ImageNet-1k |
> |-------|-------------|
> | Baseline (ViT-B) | 81.2 |
> | Proposed (ViT-B-0P) | 82.02 |
> | **Proposed (ViT-B-1P)** | **82.18** |
> | Proposed (ViT-B-3P) | 82.17 |
> | Proposed (ViT-B-Mix-Depth) | 82.16 |
>
> &nbsp;
>
> References:
>
> [1]: Stephen Zhang and Vardan Papyan: Attention Sinks: A 'Catch, Tag, Release' Mechanism for Embeddings.
>
>
> [2]: Zhao, Y., Behnia, T., Vakilian, V. & Thrampoulidis, C. (2024). Implicit geometry of next-token prediction: From language sparsity patterns to model representations.

---

> ### Author Response · Authors · 2025-11-26
>
> > ### The discussion in 2.1 and 2.2 has been heavily motivated by Figure 2. While the figure neatly conveys the idea of how a single attention block modifies the token’s relative positions, it is not clear to me that this would work as neatly in high-dimensional spaces. Some empirical support or citations for the statements made in 2.1 and 2.2 can improve the presentation.
>
> &nbsp;
>
> We thank the reviewer for raising this important point. In response, we have added Appendix D to the paper, where we empirically validate our interpretation of the geometric roles of the two head types. Figure 19 illustrates the signal to noise ratio (SNR) of token embeddings after the projection onto the sphere via Layer Normalization at each block. The SNR of a sequence is defined as the $l_2$ norm of the mean over the standard deviation of the sequence:
>
> $$
> \mathrm{SNR}(X) = \frac{\lVert \mu(X) \rVert_2}{\mathrm{std}(X)}.
> $$
>
> Thus, it directly measures how large the mean of a sequence is relative to the variance. Figure 19 shows that incorporating the Laplacian heads leads to higher SNR in the token embeddings right after projection onto the sphere. This supports two central claims made in 2.1 and 2.2: 1) attention heads collapse tokens (since the SNR increases as a function of depth) and 2) the Laplacian heads collapse tokens more efficiently.
>
> &nbsp;
>
>
> > ### While the authors give some details about training, since the method involves a different paradigm, some comparison on the training metrics with the base model (like just the number of epochs) could be useful?
>
> We thank the reviewer for this helpful suggestion. We would like to clarify that all reported experiment results were obtained using the same training recipe, hyperparameters, and training duration for both the proposed models and the baseline, including the same number of epochs and processed tokens. To improve transparency, we have expanded Table 3 in Appendix B to include more detailed descriptions of the training setup and hyperparameters.

---

> ### Author Response · Authors · 2025-11-26
>
> > ### The mixing of attention and laplacian seems a bit arbitrary. Appendix D seems to discuss more mixing trials, but I wonder if there is a way of framing this in a more principled way? Would this mixing have to be adapted at each task?
>
> We appreciate the thoughtful question from the reviewer.
>
> &nbsp;
>
> Our design choices are based on how each head type impacts the token geometry. Standard attention heads primarily move tokens in the mean direction (the radial direction of the sphere) while the Laplacian heads primarily move tokens in the variance direction (the tangent space of the sphere), as illustrated in Figure 2 of the paper.
>
> &nbsp;
>
> For the mix-depth strategy, we place attention heads earlier and Laplacian heads deeper in the network. Our intuition is that the early layers are primarily responsible for updating the token means via the standard attention heads, steering the class means towards a simplex ETF structure. On the other hand, the later layers are responsible for collapsing the tokens towards the class means via the Laplacian heads, leading to Neural Token Collapse in the final layer. Figure 4 in the paper shows indeed that later layers are those collapsing the tokens towards the means.
>
> &nbsp;
>
> For the 1P and 3P strategy, our intuition is to give each layer the flexibility to move tokens in both the mean and variance directions. Further, we note that any radial (normal) direction of the sphere is 1-dimensional while its corresponding tangent space is (d-1)-dimensional. In a C-class classification setup, each class mean corresponds to a vertex of the simplex ETF, which occupies one radial direction. Thus, one can think of the class means as occupying C dimensions, while the corresponding tangent planes are higher-dimensional. From this perspective, it is natural to use more Laplacian heads than standard attention heads.
>
> &nbsp;
>
> To investigate this further, we have added Figure 6 to the paper (converted to a table below) showing the performance of the model on ImageNet as a function of the number of P heads. The result suggests that while using the Laplacian heads only (0 P heads) is sufficient to improve upon the baseline, incorporating a small number of P heads can lead to even greater gains. This pattern corroborates with our intuition. Thus, we recommend using a small number of P heads as the default strategy.
>
>
> **Table: Top-1 ImageNet-1k accuracy (%) for different numbers $m$ of standard attention heads. $m$ = 12 is equivalent to the baseline**
>
> | $m$ | ImageNet-1k |
> |-----|-------------|
> | 0   | 82.02 |
> | 1   | **82.18** |
> | 3   | 82.17 |
> | 6   | 81.73 |
> | 9   | 81.96 |
> | 12  | 81.20 |

---

> ### Author Response · Authors · 2025-11-26
>
> > ### Some discussion on extending this framework to next token prediction/autoregressive tasks can be helpful for a more impactful paper. Do the authors think such an extension is possible, i.e., is NTC ideal for next token prediction? Some empirical validation by training small transformers using Laplacian heads would be great.
>
> &nbsp;
>
> We thank the reviewer for the detailed question. As mentioned above, we have included experimental results for autoregressive next-token prediction.
>
> &nbsp;
>
> Regarding whether token collapse is ideal for next-token prediction: At first glance, next-token prediction and token collapse appear to be at odds: autoregressive language modeling requires tokens within the same sequence to remain distinguishable, whereas full token collapse would eliminate these distinctions.
>
> &nbsp;
>
> However, recent works on attention sinks and the catch-tag-release mechanism [1] shows that tokens within the same sentence often lie in a shared low-rank subspace. When projected onto this subspace, token embeddings exhibit substantial redundancy and low-rank structure, even though they remain distinguishable in the full embedding space. This phenomenon is studied in detail in [2], which we discuss in more detail in the newly added paragraph in the Related Works section.
>
> &nbsp;
>
> This motivates a more nuanced view of collapse in language models. Rather than advocating for global token collapse, we posit that collapse restricted to task-relevant subspaces can be both compatible with — and useful for — next-token prediction. Such structured collapse may play functional roles such as organizing tokens by sentence or segmenting longer contexts into coherent units.
>
> &nbsp;
>
> From this perspective, our proposed Laplacian mechanism is not intended to induce global collapse. Instead, it aims to make this subspace-level collapse more efficient and controllable, by directly acting along the variance directions. In this way, Laplacian heads can reinforce a behavior that already exists in standard transformers, while preserving the token-level information required for autoregressive next-token prediction.

---

> > ### Comment · Reviewer_XAcD · 2025-11-27
> > **Reply to authors additions and comments**
> >
> > I thank the reviewers for replying extensively to all my concerns. I believe the additions have solved my concerns and made the manuscript more solid. I will raise my mark to accept.

---

### Official Review · Reviewer_3TuH · 2025-10-31

**Soundness:** 3
**Presentation:** 3
**Contribution:** 2
**Rating:** 4
**Confidence:** 2

**Summary:**

This work presents a method for obtaining better token embeddings, as motivated by the neural collapse phenomenon in the existing literature. Extensive empirical evidence is presented to backup the idea that current token embeddings are suboptimal, and a simple fix can be done to mitigate this. In particular, experiments on vision transformers are done on CIFAR classification tasks, as well as probing into the geometry of the existing embeddings.

**Strengths:**

- The work provides an interesting perspective on how to improve token embeddings, which can only be beneficial transformer/language model pipelines.
- Experiments showing the existing geometry of embeddings and neural token collapse are shown, as well as improvements on image classification tasks.
- The visualizations given are very useful for the reader.

**Weaknesses:**

- Experiments are run on image classification tasks only; it would be interesting to see if such improvements hold for generative models as well.

**Questions:**

- On the classification tasks, it seems that the improvement is not extremely significant. I wonder if it is strictly necessary to have better token embeddings (e.g. the ones that satisfy this analogue of neural collapse).
- Following up on my above comment (see weaknesses) for generative models: in current LLMs there are positional embeddings that encode the tokens in addition to the token embeddings themselves; is there any reason to believe that the marginal improvement towards the NTC regime will be significant enough for the case where we apply positional embeddings?

---

> ### Author Response · Authors · 2025-11-26
>
> We appreciate the reviewer for their thoughtful questions, which we address below:
>
> &nbsp;
>
> > ### Experiments are run on image classification tasks only; it would be interesting to see if such improvements hold for generative models as well.
>
> We acknowledge the reviewer’s concern. In response, we have added results for an almost-1B parameter transformer language model trained for next-token prediction on 20B FineWeb-Edu tokens. Our new results (Table 2 in paper, attached below) show that after training on the same number of tokens, the proposed modification consistently achieves better performance than the baseline on a variety of zero-shot downstream tasks. This provides evidence that the proposed modification is applicable to other domains.
>
>
>
> **Table: Zero-shot results of 836M GPT2-style decoder-only transformers on downstream datasets.**
> *Metric = accuracy (higher is better).*
>
> | Model                         | ARC-Easy | ARC-Challenge | HellaSwag | PIQA  | RACE | OpenBookQA | WinoGrande | SciQ |
> |------------------------------|----------|----------------|-----------|-------|------|-------------|-------------|------|
> | Baseline (836M)              | 63.8     | 31.48          | 36.22     | 68.55 | 30.62 | 33.2       | 53.99       | 84.1 |
> | **Proposed: 5 P-heads (836M)** | **64.56** | **32.08**      | **36.4**  | **68.71** | **32.15** | **34.6** | **54.78** | **85.2** |
>
> &nbsp;
>
> > ### On the classification tasks, it seems that the improvement is not extremely significant. I wonder if it is strictly necessary to have better token embeddings (e.g. the ones that satisfy this analogue of neural collapse).
>
> &nbsp;
>
> In response, we have updated the results in Table 1 with new ImageNet results showing larger improvements upon the baseline (attached below). Originally, our experiments showed an improvement of 0.3% to 0.6% in top-1 validation accuracy upon the baseline accuracy of 80.32%. The new results show an improvement of **0.8% to almost 1%** upon a higher **baseline of 81.2%**, which seems significant. These new results were obtained by further optimizing hyper-parameters and the training recipe, which we have detailed in Appendix B. We would like to point out that both the baseline and proposed methods share the exact same training recipe, hyper-parameters and training duration, and that the improvement comes at no extra trainable parameters. We also want to highlight that the proposed modification yields a ~5% improvement upon CIFAR100.
>
> &nbsp;
>
> > ### Following up on my above comment (see weaknesses) for generative models: in current LLMs there are positional embeddings that encode the tokens in addition to the token embeddings themselves; is there any reason to believe that the marginal improvement towards the NTC regime will be significant enough for the case where we apply positional embeddings?
>
> &nbsp;
>
> We thank the reviewer for the detailed question. We would like to clarify that by “token embeddings”, we do not mean the output of the very first embedding layer. Rather, we refer to representations that correspond to the tokens at any layer as “token embeddings”. Additionally, positional encodings were used in both the vision transformer experiments and the newly added language modelling experiment, where ViTs use learned positional encodings and the language models use the rotary positional encoding (RoPE). In all these experiments, we see that the proposed modification yields noticeable improvements. Hence, we believe the benefit of the NTC regime is significant when different types of positional encodings are applied.

---

### Official Review · Reviewer_5VSE · 2025-11-03

**Soundness:** 2
**Presentation:** 2
**Contribution:** 2
**Rating:** 4
**Confidence:** 2

**Summary:**

This paper proposes the concept of Neural Token Collapse (NTC) as a desirable geometric property for token embeddings in transformers. They show that standard transformer architectures fail to achieve the so called ideal token geometry. To address this, the authors introduce a a simple modification to the standard attention mechanism where the output PV is changed to V−PV. The paper demonstrates through experiments on CIFAR-10, CIFAR-100, and ImageNet that this modification not only brings the token geometry closer to the ideal NTC but also yields consistent and significant improvements in classification accuracy.

**Strengths:**

-The proposed idea seems to be novel and simple. It simply modified the attention output from PV to V - PV. It offers a practical and low-effort way to improve existing transformer models without adding parameters or significant computational cost.

-The proposed method achieves meaningful performance improvements across all tested datasets, including a 5% absolute improvement on CIFAR-100.

-Interesting theoretical insights. The paper provides a compelling theoretical interpretation of the proposed change by connecting it to graph theory.

**Weaknesses:**

- It is unclear that why the proposed NTC is the ideal token geometry.

- Limited Scope of evaluation. All experiments are conducted on image classification. While the theory is general, the paper lacks evidence of its applicability to other domains, most notably Natural Language Processing. It is an open question whether forcing this kind of NTC would be beneficial for other tasks.

- The paper proposes mixing standard Attn heads with the new Laplacian L heads to further improve performance. However, the strategies for this mixing (e.g., "1P", "3P", "Mix-Depth") feel somewhat heuristic and are not as well-motivated as the core Laplacian idea itself.

**Questions:**

The authors are encouraged to provide evaluation results in NLP.

---

> ### Author Response · Authors · 2025-11-26
>
> We would like to thank the reviewer for their constructive feedback and for finding our work interesting. Below, we attempt to address each of the reviewer's concerns:
>
> > ### Limited Scope of evaluation. All experiments are conducted on image classification. While the theory is general, the paper lacks evidence of its applicability to other domains, most notably Natural Language Processing. It is an open question whether forcing this kind of NTC would be beneficial for other tasks.
>
> &nbsp;
>
> We acknowledge the reviewer’s concern. In response, we have added results for an almost-1B parameter transformer language model trained for next-token prediction on 20B FineWeb-Edu tokens. Our new results (Table 2 in paper, attached below) show that after training on the same number of tokens, the proposed modification consistently achieves better performance than the baseline on a variety of zero-shot downstream tasks. This provides evidence that the proposed modification is applicable to other domains.
>
>
> **Table: Zero-shot results of 836M GPT2-style decoder-only transformers on downstream datasets.**
> *Metric = accuracy (higher is better).*
>
> | Model                         | ARC-Easy | ARC-Challenge | HellaSwag | PIQA  | RACE | OpenBookQA | WinoGrande | SciQ |
> |------------------------------|----------|----------------|-----------|-------|------|-------------|-------------|------|
> | Baseline (836M)              | 63.8     | 31.48          | 36.22     | 68.55 | 30.62 | 33.2       | 53.99       | 84.1 |
> | **Proposed: 5 P-heads (836M)** | **64.56** | **32.08**      | **36.4**  | **68.71** | **32.15** | **34.6** | **54.78** | **85.2** |
>
>
> &nbsp;
>
> > ### It is unclear why the proposed NTC is the ideal token geometry.
>
> &nbsp;
>
> We would like to clarify that NTC is a conjectured ideal token geometry motivated by two lines of prior work: 1) rank collapse, and 2) Neural Collapse (NC). Works on rank collapse suggest that transformers have an inherent inductive bias towards collapsing tokens to single points, while works on Neural Collapse suggest that the collapsed points should align with a simplex ETF. Based on these works, we hypothesized that Neural Token Collapse is the ideal geometry for classification via transformers. Our work empirically supports this conjecture by showing a correlation between the emergence of Neural Token Collapse and better classification performance. To show a stronger correlation, we have updated Table 1 with new ImageNet results showing larger improvements upon the baseline (attached below).
>
>
> **Table: Top-1 ImageNet-1k accuracy (%) of models.**
>
> | Model | ImageNet-1k |
> |-------|-------------|
> | Baseline (ViT-B) | 81.2 |
> | Proposed (ViT-B-0P) | 82.02 |
> | **Proposed (ViT-B-1P)** | **82.18** |
> | Proposed (ViT-B-3P) | 82.17 |
> | Proposed (ViT-B-Mix-Depth) | 82.16 |

---

> ### Author Response · Authors · 2025-11-26
>
> > ### The paper proposes mixing standard Attn heads with the new Laplacian L heads to further improve performance. However, the strategies for this mixing (e.g., "1P", "3P", "Mix-Depth") feel somewhat heuristic and are not as well-motivated as the core Laplacian idea itself.
>
> &nbsp;
>
> We appreciate the astute question from the reviewer.
>
> &nbsp;
>
> Our design choices are based on how each head type impacts the token geometry. Standard attention heads primarily move tokens in the mean direction (the radial direction of the sphere) while the Laplacian heads primarily move tokens in the variance direction (the tangent space of the sphere), as illustrated in Figure 2 of the paper.
>
> &nbsp;
>
> For the mix-depth strategy, we place attention heads earlier and Laplacian heads deeper in the network. Our intuition is that the early layers are primarily responsible for updating the token means via the standard attention heads, steering the class means towards a simplex ETF structure. On the other hand, the later layers are responsible for collapsing the tokens towards the class means via the Laplacian heads, leading to Neural Token Collapse in the final layer. Figure 4 in the paper shows indeed that later layers are those collapsing the tokens towards the means.
>
> &nbsp;
>
> For the 1P and 3P strategy, our intuition is to give each layer the flexibility to move tokens in both the mean and variance directions. Further, we note that any radial (normal) direction of the sphere is 1-dimensional while its corresponding tangent space is (d-1)-dimensional. In a C-class classification setup, each class mean corresponds to a vertex of the simplex ETF, which occupies one radial direction. Thus, one can think of the class means as occupying C dimensions, while the corresponding tangent planes are higher-dimensional. From this perspective, it is natural to use more Laplacian heads than standard attention heads.
>
> &nbsp;
>
> To investigate this further, we have added Figure 6 to the paper (converted to a table below) showing the performance of the model on ImageNet as a function of the number of P heads. The result suggests that while using the Laplacian heads only (0 P heads) is sufficient to improve upon the baseline, incorporating a small number of P heads can lead to even greater gains. This pattern corroborates with our intuition. Thus, we recommend using a small number of P heads as the default strategy.
>
>
> **Table: Top-1 ImageNet-1k accuracy (%) for different numbers $m$ of standard attention heads. $m$ = 12 is equivalent to the baseline**
>
> | $m$ | ImageNet-1k |
> |-----|-------------|
> | 0   | 82.02 |
> | 1   | **82.18** |
> | 3   | 82.17 |
> | 6   | 81.73 |
> | 9   | 81.96 |
> | 12  | 81.20 |

---

### Official Review · Reviewer_aBvF · 2025-11-03

**Soundness:** 2
**Presentation:** 3
**Contribution:** 1
**Rating:** 2
**Confidence:** 4

**Summary:**

This paper proposes neural token collapse (NTC) as an ideal token geometry for transformers in the setting of classification. Inspired by the neural collapse line of work, this work focuses on the specific case of transformer architectures and proposes simple architectural fixes to the attention mechanism to promote NTC. Experiments are conducted on image classification tasks to verify the effectiveness of the proposed method.

**Strengths:**

1. The proposed method is very simple and can be incorporated into standard transformers easily.
2. The geometric framing is straightforward and the ANOVA approach makes the results easy to analyze and interpret.
3. The connection to diffusion over graphs is interesting and aligns well with the experimental results.

**Weaknesses:**

1. It is very questionable whether achieving zero variance among tokens within the same sequence is truly desirable. Note self-attention precisely promotes dynamic weighting between tokens as a mechanism to propagate information. Removing the variance among tokens leads to a trivial weighting and effectively makes self-attention no better than a mean aggregation. To test this, one can add another baseline with just a simple mean operator to perform token mixing, without any self-attention.
2. The method only makes sense for classification, and I find it hard to extend the method to the vast amount of tasks transformers do well in: causal language modeling, dense segmentation, masked image modeling and other self-supervised visual modeling.
3. In experiments, the authors only tested the model with the hybrid approach described in Sec.2.3. It is not clear what the performance would be if using only the proposed method in Sec 2.2. This makes it hard to understand the behavior of the proposed method.
4. Related to the previous point, the results are not truly convincing, as the classification performance gain on ImageNet-1k is very small. It is questionable whether the proposed fix alone is worth incorporating into any existing ViTs.

**Questions:**

Please see above.

---

> ### Author Response · Authors · 2025-11-26
>
> We appreciate the reviewer’s feedback for our work. We will attempt to address each of the reviewer’s concerns below.
>
> &nbsp;
>
> > ### The method only makes sense for classification, and I find it hard to extend the method to the vast amount of tasks transformers do well in: causal language modeling, dense segmentation, masked image modeling and other self-supervised visual modeling.
>
> &nbsp;
>
> We acknowledge the reviewer’s concern. In response, we have added results for an almost-1B parameter transformer language model trained for next-token prediction on 20B FineWeb-Edu tokens. Our new results (Table 2 in paper, attached below) show that after training on the same number of tokens, the proposed modification consistently achieves better performance than the baseline on a variety of zero-shot downstream tasks. This provides evidence that the proposed modification is applicable to other domains.
>
>
> **Table: Zero-shot results of 836M GPT2-style decoder-only transformers on downstream datasets.**
> *Metric = accuracy (higher is better).*
>
> | Model                         | ARC-Easy | ARC-Challenge | HellaSwag | PIQA  | RACE | OpenBookQA | WinoGrande | SciQ |
> |------------------------------|----------|----------------|-----------|-------|------|-------------|-------------|------|
> | Baseline (836M)              | 63.8     | 31.48          | 36.22     | 68.55 | 30.62 | 33.2       | 53.99       | 84.1 |
> | **Proposed: 5 P-heads (836M)** | **64.56** | **32.08**      | **36.4**  | **68.71** | **32.15** | **34.6** | **54.78** | **85.2** |
>
>
> &nbsp;
>
>
> > ### The results are not truly convincing, as the classification performance gain on ImageNet-1k is very small. It is questionable whether the proposed fix alone is worth incorporating into any existing ViTs.
>
>
> &nbsp;
>
>
> In response, we have updated Table 1 with new ImageNet-1k results (attached below) that show greater improvements upon the baseline. Originally, our experiments showed an improvement of 0.3% to 0.6% in top-1 validation accuracy upon a baseline accuracy of 80.32%. The new results show an improvement of **0.8% to 1% upon a higher baseline of 81.2%**. These new results are obtained by further optimizing the hyper-parameters and training recipe for the baseline, which we have detailed in Appendix B. We would like to emphasize that the improvement comes at no extra trainable parameters and the same hyperparameters used for the baseline are also used for the proposed models. Given the simplicity of the method, we hope these results validate the potential of incorporating our method into existing ViTs.
>
>
> **Table: Top-1 ImageNet-1k accuracy (%) of models.**
>
> | Model | ImageNet-1k |
> |-------|-------------|
> | Baseline (ViT-B) | 81.2 |
> | Proposed (ViT-B-0P) | 82.02 |
> | **Proposed (ViT-B-1P)** | **82.18** |
> | Proposed (ViT-B-3P) | 82.17 |
> | Proposed (ViT-B-Mix-Depth) | 82.16 |
>
> &nbsp;
>
> > ### In experiments, the authors only tested the model with the hybrid approach described in Sec.2.3. It is not clear what the performance would be if using only the proposed method in Sec 2.2. This makes it hard to understand the behavior of the proposed method.
>
> &nbsp;
>
> We acknowledge the reviewer's valid concern. In response, we have now added results for only using the Laplacian heads for all image datasets and metrics. The new results show that only using the Laplacian heads leads to a similar impact on representation geometry as the other proposed models. Additionally, we have added Figure 6 to the paper (converted to a table below) showing the model’s performance on ImageNet as a function of the number of P heads. These results suggest that using the Laplacian heads only (0 P heads) achieves consistent improvements upon the baseline and leads to more prominent Neural Token Collapse in the token embeddings. However, we also find that incorporating a **small** number of P heads can sometimes achieve even greater gains, and it never harms performance, which is why this strategy is our default recommendation.
>
> **Table: Top-1 ImageNet-1k accuracy (%) for different numbers $m$ of standard attention heads. $m$ = 12 is equivalent to the baseline**
>
> | $m$ | ImageNet-1k |
> |-----|-------------|
> | 0   | 82.02 |
> | 1   | **82.18** |
> | 3   | 82.17 |
> | 6   | 81.73 |
> | 9   | 81.96 |
> | 12  | 81.20 |

---

> ### Author Response · Authors · 2025-11-26
>
> > ### It is very questionable whether achieving zero variance among tokens within the same sequence is truly desirable.
>
> &nbsp;
>
> We thank the reviewer for raising this important question. To address this concern more clearly, we have added a new paragraph in Section 7 of the paper briefly discussing this issue, which we expand further here.
>
> &nbsp;
>
> Firstly, our method does not inherently enforce token collapse. The Laplacian head provides the model with additional flexibility to control intra-sequence token variance. In particular, it can move tokens **either toward or away from** the sequence mean, depending on what is favored by the task and training objective. Thus, the mechanism does not necessarily prescribe zero variance.
>
> &nbsp;
>
> Secondly, our work focuses on **supervised classification**, where we observe that the emergence of Neural Token Collapse correlates with improved performance. We do not claim that zero variance is universally optimal, and we explicitly leave the question of optimal token geometry for other tasks as an open direction for future work.
>
> &nbsp;
>
> Finally, recent work on attention sinks and the catch–tag–release mechanism [1] has reported that token embeddings within a sequence in large language models are often approximately **low-rank**, and exhibit near-zero variance **after projection onto relevant subspaces**. Thus, by allowing tokens to move more freely along their variance directions **within the relevant subspaces**, the Laplacian heads could help promote geometric structures that already exist in current models.
>
> &nbsp;
>
> > ### Note self-attention precisely promotes dynamic weighting between tokens as a mechanism to propagate information. Removing the variance among tokens leads to a trivial weighting and effectively makes self-attention no better than a mean aggregation. To test this, one can add another baseline with just a simple mean operator to perform token mixing, without any self-attention.
>
> &nbsp;
>
> We appreciate the reviewer’s observation. It is indeed correct that if all variance among tokens is removed, then self-attention becomes a trivial mean aggregation. However, this situation never arises in our setup. The NTC geometry only emerges at the final layer of the model, while in all earlier layers, tokens retain a significant amount of variance, as shown in the layer-wise cosine similarity measurement (Figure 4 in the paper). Thus, in all of our experiments, neither self-attention nor the Laplacian mechanism is equivalent to the trivial mean aggregation.
>
> &nbsp;
>
> [1] Stephen Zhang and Vardan Papyan: Attention Sinks: A 'Catch, Tag, Release' Mechanism for Embeddings.
>
> &nbsp;
>
> We would like to thank the reviewer for their thoughtful critique of our work. Should the reviewer find our responses satisfactory, we would always appreciate it if the reviewer would reconsider the score.

---

### Author Response · Authors · 2025-12-03
**Summary of Revision**

We would like to thank all reviewers again for their thoughtful questions and feedback. We have carefully responded to each concern/question and revised the paper accordingly. The revisions we made are highlighted in blue in the updated paper. Below, we summarize all substantial revisions:

&nbsp;

### Experiment on Language

We added results for transformer language models trained for next-token prediction on 20B FineWeb-Edu tokens (Section 4.6.2). We included full details of the language modelling experiments in Appendix B2.

&nbsp;

### Better ImageNet Results

We updated the ImageNet experiments showing a larger improvement upon a higher baseline (Table 1). The results were obtained by further optimizing the training recipe, which we detail in Appendix B1.

&nbsp;

### Results for Only Using the Laplacian Heads

We added experiment results for only using the Laplacian heads (denoted ViT-B-0P in the paper). This includes image classification results for all datasets and all proposed metrics.

&nbsp;

### Empirical Justification for Mixing Strategy

We added a figure showing the performance on ImageNet as a function of the number of standard attention heads (Figure 6) to justify the mixing strategy defined in Section 2.3. We also provided further details about these experiments in Appendix E (Table 5).

&nbsp;

### Empirical Evidence for Geometric Interpretations

We added Appendix D showing empirical evidence for our geometric interpretation (Figure 2) of the standard attention and the proposed Laplacian mechanisms.

&nbsp;

### New Paragraph in Related Works

We added a new paragraph in Section 6 discussing related works that study token collapse in autoregressive language models.

---

### Author Response · Authors · 2025-12-03
**Rebuttal Summary for AC**

Thank you for reviewing our work. During the rebuttal, we have tried our best to address every concern/question raised by the reviewers. We believe we have addressed all concerns adequately. For example, Reviewer XAcD raised their score from 6 to 8 before the rebuttal was interrupted. If the rebuttal had continued, we expect that the other reviewers would similarly have raised their scores.

Here we summarize the reviews and our attempt to address the reviewers’ concerns.

## Strengths

All reviewers agreed that the Laplacian mechanism we proposed was novel and simple to incorporate into existing transformers. They agreed that our analysis of how the proposed modification impacts representation geometry is interesting and clear. Finally, reviewers also found our method well-motivated and compelling due to its connection to diffusion over graphs.

&nbsp;

## Reviewers’ Concerns and Questions

&nbsp;

### Missing Language Modelling Experiments

All reviewers unanimously asked for experiment results on causal language modelling. In response, we have provided experiment results for training transformer models with almost 1B parameters on 20B FineWeb-Edu tokens. These results show that the proposed modification leads to better performance on a variety of zero-shot downstream tasks (Section 4.6.2).

&nbsp;

### Improvements Not Significant Enough

Reviewers aBvF and 3TuH questioned whether the improvements achieved by our method are significant enough. In response, we updated Table 1 (Section 4.6.1) with new ImageNet results showing more significant improvements of our method upon a higher baseline. The old results show an improvement of 0.3% to 0.6% in top-1 validation accuracy upon a baseline accuracy of 80.32% while the new results show an improvement of 0.8% to almost 1% upon a higher baseline of 81.2%.

&nbsp;

### Missing Results for Only Using the Laplacian Heads

Reviewer aBvF requested experiment results for using the proposed Laplacian mechanism for all heads. In response, we provided results for this option (denoted ViT-B-0P in the paper) on all datasets and metrics (Table 1 and Appendix C).

&nbsp;

### Mixing Strategies Not Well-Motivated Enough

Reviewers 5VSE and XAcD commented that the proposed strategies for mixing standard attention heads and the Laplacian heads felt arbitrary. In response, we provided experiment results showing ImageNet performance as a function of the number of standard attention heads (Figure 6), thus empirically justifying our strategies. We also provided a detailed discussion of the mathematical motivation behind this idea (See response to these two reviewers).

&nbsp;

### Is our geometric interpretation of attention and the Laplacian valid?

Reviewer XAcD questioned whether our interpretation of how standard attention and the Laplacian mechanism collapse tokens is valid in high dimensions (Figure 2). In response, we added Appendix D containing empirical evidence supporting our geometric interpretations.

&nbsp;

### Is Token Collapse desirable? Is it desirable for next-token prediction?

Reviewer aBvF expressed skepticism about whether token collapse is desirable. In our response, we clarified in great detail that:
1. The proposed Laplacian head does not inherently enforce token collapse.
2. Our work focuses exclusively on supervised classification, and we do not claim token collapse to be universally optimal.
3. Some recent related works show that collapse within certain subspaces could be useful for causal language modelling. Thus, the Laplacian heads could be beneficial for this task.

We added a paragraph at the end of the paper briefly discussing 1 and 2.

Related to 3, Reviewer XAcD asked whether token collapse is beneficial for next-token prediction. We provided a detailed discussion of this question in our response. We also added a paragraph in the Related Works section (Section 6) discussing this question.

---

### Meta-Review · Area_Chair_CVHw · 2026-01-06

**Summary:**

Initial reviews are somewhat negative (2,4,4,8).

Reviewer aBvF questions the premise that the optimal geometric arrangement of tokens should be to collapse to single points and whether the proposed model makes sense for non-classification tasks along with questions on the demonstrated performance gains.

Reviewer 5VSE also questions the target token geometry and evaluation setting.  The reviewer also questions some of the mixing strategies and suggests they may not be well motivated by the core Laplacian framework.

Reviewer 3TuH likewise questions whether performance improvements can be observed beyond classification tasks and notes that on the presented experiments the gains appear relatively minor.

Reviewer XAcD also notes some weaknesses in the experimental performance and presentation.

**Reviewer Concerns:**

Overall, the authors have made a strong attempt to address the reviewers concerns by adding experiments on language modeling and further tuning model hyperparameters to improve the model and baseline performance.

In addition, the authors also discuss how their approach doesn't necessarily enforce token collapse but allows the model to control token variance.

However, in both cases, I am not sure the rebuttal adequately addresses the issue.  For example, in the case of token collapse, while it is true that this may not be strictly enforced, the proposed operator in (3) would appear to be applying a smoothing of the token signals over the graph defined in P (as a Laplacian operator does) and thus be encouraging movement towards such a geometry.

Moreover, given the relatively subtle change in the resulting operator over standard self-attention it is perhaps not surprising that the proposed model performs rather similarly with a slight increase in performance.

**Reviewer Scores:**

Given the relatively low initial scores from the reviewers, it is not clear to me that the rebuttal would be sufficient to significantly sway the reviewers' opinion to the degree required to arrive at a consensus for acceptance.

While this authors have made commendable improvements to the work with the additional experimental evaluation, I am not convinced that this would necessarily resolve the concerns raised by the reviewers.  I would encourage the authors to consider the reviewers' feedback in preparing a revised version to better motivate the proposed method and demonstrate it's practical effectiveness.

---

### Decision · Program_Chairs · 2026-01-26

Reject